



# On the similarity and apparent cycles of isotopic variations in East Antarctic snow-pits

Thomas Laepple[1], Thomas Münch[1,2], Mathieu Casado[3], Maria Hoerhold[4], Amaelle Landais[3], and Sepp Kipfstuhl[4]

[1]Alfred Wegener Institute Helmholtz Centre for Polar and Marine Research, Telegrafenberg A43, 14473 Potsdam, Germany
[2]Institute of Physics and Astronomy, University of Potsdam, Karl-Liebknecht-Str. 24/25, 14476 Potsdam, Germany
[3]Laboratoire des Sciences du Climat et de l'Environnement - IPSL, UMR 8212, CEA-CNRS-UVSQ, Gif sur Yvette, France
[4]Alfred Wegener Institute Helmholtz Centre for Polar and Marine Research, Am Alten Hafen 26, 27568 Bremerhaven, Germany

*Correspondence to:* Thomas Laepple (thomas.laepple@awi.de)

**Abstract.** Stable water isotopes in polar ice provide a wealth of information about past climate evolution. Snow pit studies allow us to relate observed weather and climate conditions to the measured isotope variations in the snow. They therefore offer the possibility to test our understanding of how isotope signals are formed and stored in firn and ice. As stable water isotopes in the snowfall are strongly correlated to air temperature, isotopes in the near surface snow are supposed to depict the seasonal cycle at a given site. Accordingly, variation between sites in accumulation rate is expected to be matched by variation in the number of seasonal cycles. However, snowpit studies from different accumulation conditions in East Antarctica reported similar isotopic variability and comparable apparent cycles in the $\delta^{18}$O and $\delta$D profiles with typical wavelengths of $\sim 20$ cm. These observations are unexpected as the accumulation rates strongly differ between the sites, ranging from 20 to 80 mm w.e. yr$^{-1}$ ($\sim 5$–25 cm of snow per year). Various mechanism have been proposed to explain the isotopic variations individually at each site; however, none of these is consistent with the similarity of the different profiles independent of the local accumulation conditions.

Here, we systematically analyze the properties and origins of isotopic variations in high-resolution firn profiles from eight East Antarctic sites. First, we confirm the suggested cycle length (mean distance between peaks) of $\sim 20$ cm by counting the isotopic maxima. Spectral analysis further shows a strong similarity between the sites but indicates no dominant periodic features. Finally, the apparent cycle length increases with depth for most East Antarctic sites, which is inconsistent with burial and compression of a regular seasonal cycle. We show that these results can be explained by isotopic diffusion acting on a noise dominated isotope signal. The firn diffusion length is rather stable across the East Antarctic and thus leads to similarpower spectral densities of the isotopic variations. This in turn implies a similar distance between isotopic maxima in the firn profiles.

Our results explain a large set of observations discussed in the literature, providing a simple explanation for the interpretation of apparent cycles in shallow isotope records, without invoking complex mechanisms. Finally, the results underline previous suggestions that isotopic signals in single ice cores from low-accumulation regions have a small signal-to-noise ratio and thus likely do not allow the reconstruction of interannual to decadal climate variations.



# 1 Introduction

Stable water isotope records from ice cores can be used to infer past local temperature variations (Dansgaard, 1964) and as such are an important climate proxy at interannual to glacial-interglacial timescales (Jouzel et al., 2007; Johnsen et al., 2001).The depth and accumulation rate of an ice core affects the temporal scale and resolution of the climate reconstruction that can be

obtained. In central East Antarctica, low accumulation rates and deep ice cores allow climate reconstructions to be made that cover the last 800 000 years (EPICA community members, 2004), while the higher accumulation rates in coastal areas allow for higher resolution reconstructions in which the seasonal climate signal can be recovered from the ice isotopic composition (Morgan, 1985; Masson-Delmotte et al., 2003; van Ommen and Morgan, 1997; Küttel et al., 2012).

High-resolution isotope data, thought to correspond to sub-seasonal variations are now routinely measured at deep ice core

sites (Gkinis et al., 2011); however, it is unclear to which extent isotopic signals on timescales shorter than multidecadal can be interpreted as indicating climate (Ekaykin et al., 2002; Baroni et al., 2011; Pol et al., 2014; Münch et al., 2016), especially for the low-accumulation regions that are typical on the East Antarctic Plateau ($< 100 \, \mathrm{mm \, w.e. \, yr^{-1}}$). As the link is complex between isotopic composition and the climatic conditions creating them (Jouzel et al., 1997), numerous studies have sampled the upper meters of firn in order to compare the isotopic variations with instrumental climate data (Masson-Delmotte et al.,

2008; Fernandoy et al., 2010; Steen-Larsen et al., 2014). Many of these have reported oscillations in snow pit records from the East Antarctic Plateau, including at Vostok, Dome C (EDC), Dome A, Dome F (DF), South Pole (SP) and Kohnen Station at the the EPICA Dronning Maud Land drilling site (EDML) (Jouzel et al., 1983; Petit et al., 1982; Ekaykin et al., 2002; Hoshina et al., 2014, 2016; Münch et al., 2016). Interestingly, despite the very different annual snow layer thicknesses at these sites, which range from 6 cm to 21 cm, their isotope profiles appear to have very similar peak-to-peak distances (Fig 1) and a recent

systematic counting effort of isotopic maxima in firn profiles (Casado et al., 2016) suggested a characteristic wavelength of 15–25 cm across all analysed East-Antarctic sites.

For sites such as EDML and South Pole, their apparent cycle lengths match well with their annual snow layer thicknesses and consequently their cycles have been explained as reflecting seasonal climate variation (Oerter et al., 2004; Münch et al., 2016; Jouzel et al., 1983; Whitlow et al., 1992). However, this explanation is not consistent with the same cycle length being

observed at lower accumulation sites, where the annual snow layer thickness is often less than 10 cm (Petit et al., 1982). Instead, a range of alternative explanations have been proposed for the oscillations at individual sites. At Vostok, Ekaykin et al. (2002) attributed the oscillations to horizontally moving dunes (Frezzotti et al., 2002) leading to isotopic cycles during burial. However, similar cycles are found at core sites with different dune features, wind speeds and accumulation rates, and thus a varying speed of dune movement and burial. At EDC, Petit et al. (1982) explained the mismatch between seasonal

and isotopic cycles as being due to missing years resulting from the combined effects of successive precipitation free months, erosion associated with blowing snow, and firn diffusion. In a multi-site study, Hoshina et al. (2014, 2016) suggested that the multi-year oscillations could be formed by the combination of variable accumulation and post-depositional modifications, such as ventilation. Finally, the coarse sampling resolution in some studies (5 cm or longer) would not resolve the seasonal cycle, but similar characteristics were found for profiles sampled at a range of resolutions (Ekaykin et al., 2002). Thus, none of the




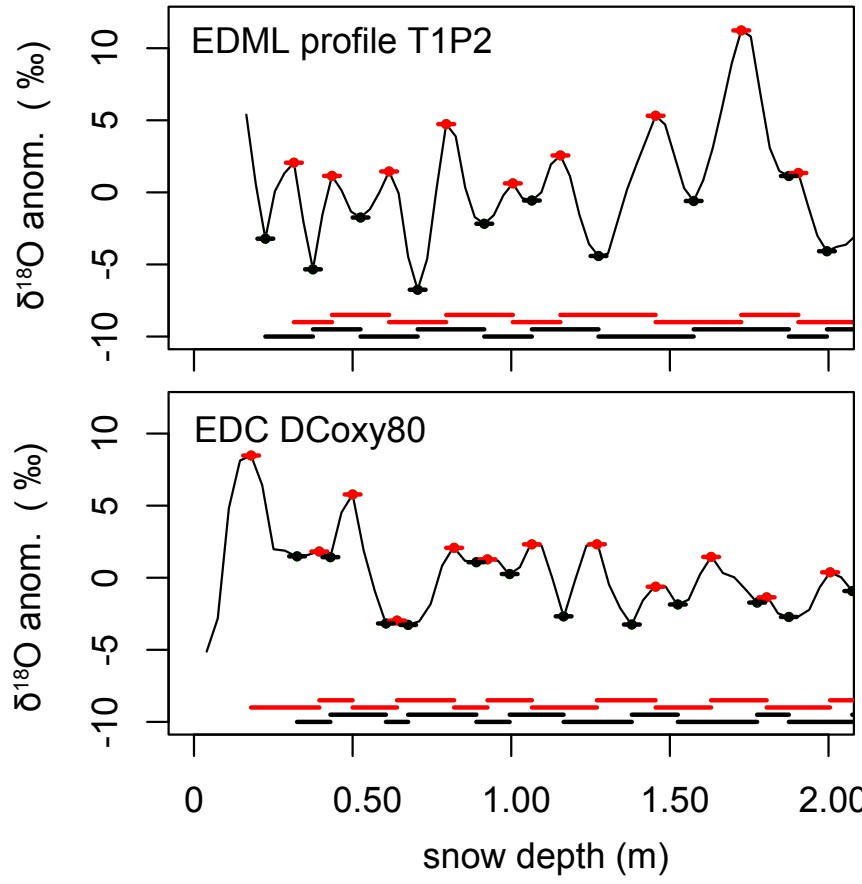

**Figure 1.** Example isotope profiles from EDML and EDC. Both profiles are visually similar despite the differing time periods covered, ∼10 yr for EDML and ∼25 yr for EDC, and annual snow layer thicknesses, ∼21 cm for EDML and ∼8 cm for EDC. Also shown is an example of the automatic estimation of isotopic cycle length. Red and black dots show the identified maxima and minima. The short horizontal lines underneath these dots indicate the ±6 cm region a maximum (minimum) must be above (below) to be identified as an extremum. The longer black and red horizontal lines at the bottom of the figure indicate the identified distances between subsequent maxima or minima.

existing interpretations explain why the apparent observed cycles are so similar across sites and largely independent of the accumulation rate and related climatic conditions.

    Here we combine a statistical analysis of isotope profiles from eight East Antarctic Plateau sites with theoretical considerations and numerical simulations of the firn signal. We suggest that the presence of apparent cycles in the firn, and their largely



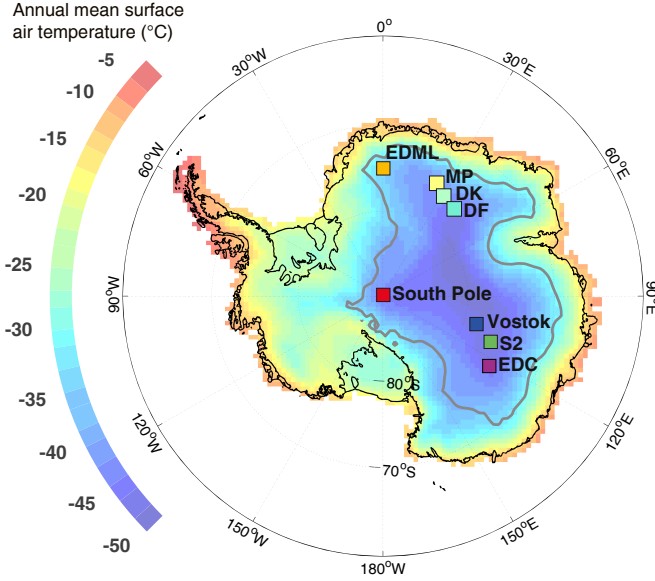

**Figure 2.** Location of the sampling sites used in this study (solid squares). The 2500 m a.s.l., elevation contour is marked by a grey line, colours indicate the annual mean surface air temperature (Nicolas and Bromwich, 2014).

invariant length, can be explained by a combination of deposition-related noise in the surface isotope signal and isotopic diffusion (Johnsen et al., 2000), which is rather constant across the East Antarctic Plateau.

## 2  Data and Methods

We first introduce the dataset and the method used to compare the power spectral density of observed isotope profiles with
5    those from a null model of diffused noise. We then provide an analytic solution for the expected distance between isotopic maxima ('cycle length') and a method to estimate this cycle length from the observed firn profiles. Finally we provide a minimal numerical forward model for the isotopic variations.

### 2.1  Data

We analyse data from eight sites on the East Antarctic Plateau, for which vertical isotope profiles of $\delta^{18}$O or $\delta$D are available
10   with lengths of at least 2 m and minimum resolutions of 3.5 cm per sample (Fig. 2 and Table 1). We focus our analysis on the upper 4 m of firn within which cycles have been described and interpreted. For the South Pole and EDML, a combination of snow pit and shallow firn core data allows us to extend the analysis down to a snow depth of 18 m. All records have been published (Table 1), except those for EDML for which we use 22 3.4 m deep profiles sampled from snow trenches (T15) as described and partly analysed in Münch et al. (2017), together with new isotope data from the firn cores B41 and B50. These





two cores were drilled close to Kohnen Station in 2012/13, approximately 1 km apart (Alfred-Wegener-Institut Helmholtz-Zentrum für Polar- und Meeresforschung, 2016). Isotope ratios were analysed at a resolution of $3\,\mathrm{cm}$, using a laser instrument at the Alfred Wegener Institute (AWI) in Bremerhaven and following the protocol described in Masson-Delmotte et al. (2015).

## 2.2 Spectral analysis

Spectra are estimated using Thomson's multitaper method with three windows (Percival and Walden, 1993). The depth profiles are linearly detrended before analysis. For profiles with non-equidistant sampling, the data is interpolated to the lowest resolution after low-pass filtering to avoid aliasing effects as described in Laepple and Huybers (2013). In the case of multiple profiles at a single site we show the mean of the individual spectra.

Significance testing of the power spectral density is performed against a null hypothesis of noise affected by firn diffusion.
More specifically, we assume the sum of white (temporally independent) noise subject to isotopic diffusion with a depth varying diffusion length and additive white measurement noise. The modeling of the isotopic diffusion and the site specific parameters are described later in detail.

There is no simple closed form expression for the power spectrum of a diffused signal under varying diffusion lengths. We thus resort to a Monte Carlo procedure by simulating diffused white noise profiles on a 2 mm resolution, resampling them to
the actual resolution of the observed profiles to include the effect of discrete sampling, adding measurement noise and then estimating the spectra on these surrogate datasets using the same method as for the observed profiles. The variance of the diffused white noise signal and the measurement noise are free parameters and are chosen to minimize the root mean square deviation of the observed and the null-hypothesis spectrum.

Finally, we scale the null-hypothesis spectrum by the 95% percent quantiles of a $\chi^2$ distribution to obtain critical significance
levels. These mark the range of spectral power expected if the time-series were diffused noise. This level has to be overcome for a peak to be locally (thus at a given frequency) significant. The degrees of freedom (DOF) of the $\chi^2$ distribution are the product of the DOF from the spectral estimator and the effective number of independent profiles. We assume independence between profiles for all sites except for the EDML trench data. For this dataset consisting of 22 nearby profiles that were sampled in a single season, we assume five effective degrees of freedom.

## 2.3 Rice's formula for the expected number of local extrema

The 'wiggliness' of time series, assuming a stationary random process, is determined by the first moments of the spectral density. This relationship known as Rice's formula (Rice, 1944, 1945) was shown to have important implications for interpreting paleoclimate records (Wunsch, 2006) and can be used to derive the expected number $\mu$ of local extremes (maxima and minima) per unit time. Specifically, for a stationary Gaussian process (e.g. Lindgren, 2012), $\mu$ is

$$\mu_{\max} = \mu_{\min} = \frac{1}{2\pi}\sqrt{\frac{\Omega_4}{\Omega_2}}, \tag{1}$$

where $\Omega_2$ and $\Omega_4$ denote the second and the fourth moment of the spectral density of the Gaussian process.





**Table 1.** Summary of the drilling and snow pit sites used in this study. For each site we list latitude, longitude and elevation above sea level; for each corresponding record its name, the analysed depth, sampling resolution, measured proxy and original data reference. Some profiles have single missing measurements that are not included in the provided sampling resolution. The 35 missing $\delta^{18}O$ samples in DCoxy80 were filled by linear regression to $\delta D$. For SP, $\delta D$ was converted to $\delta^{18}O$ using a slope of 8 to get a complete $\delta^{18}O$ dataset

| Site | Lat. °N | Lon. °E | Elevation m a.s.l. | Name | Depth m | Resolution cm | Proxy | Reference |
|---|---|---|---|---|---|---|---|---|
| Vostok | −78.5 | 106.8 | 3488 | VK14 | 2.5 | 2 | $\delta^{18}O, \delta D$ | Ekaykin et al. (2002) |
| | | | | ST61 | 3.1 | 2 | $\delta^{18}O, \delta D$ | Ekaykin et al. (2002) |
| | | | | ST73 | 3.1 | 2 | $\delta^{18}O, \delta D$ | Ekaykin et al. (2002) |
| | | | | ST30 | 3 | 2.0–3.5 | $\delta^{18}O, \delta D$ | Ekaykin et al. (2002) |
| | | | | VK56 | 3 | 3 | $\delta^{18}O, \delta D$ | Touzeau et al. (2016) |
| EDC | −75.1 | 123.3 | 3233 | DCoxy5 | 2 | 1–2 | $\delta^{18}O, \delta D$[†] | Casado et al. (2016) |
| | | | | DCoxy80 | 2.5 | 3.3–3.6 | $\delta^{18}O, \delta D$ | Casado et al. (2016) |
| | | | | Vanish12 | 2 | 3 | $\delta^{18}O, \delta D$ | Touzeau et al. (2016) |
| DF | −77.3 | 39.7 | 3810 | DF | 4 | 2 | $\delta^{18}O, \delta D$ | Hoshina et al. (2014) |
| DK | −76.8 | 31.8 | 3733 | DK | 2 | 2 | $\delta^{18}O, \delta D$ | Hoshina et al. (2014) |
| S2 | −76.3 | 120.0 | 3229 | S2 | 3 | 2.9–3.0 | $\delta^{18}O, \delta D$ | Touzeau et al. (2016) |
| MP | −74.0 | 43.0 | 3656 | MP | 4 | 2 | $\delta^{18}O, \delta D$ | Hoshina et al. (2014) |
| EDML | −75.0 | 0.1 | 2892 | T15 (22 profiles) | 3.4 | 2.2–3.0 | $\delta^{18}O, \delta D$ | Münch et al. (2017) + this study |
| | | | | B41 | 12 * | 2 | $\delta^{18}O, \delta D$ | this study |
| | | | | B50 | 12 * | 2 | $\delta^{18}O, \delta D$ | this study |
| SP | −90.0 | 0.0 | 2835 | SP78P | 10 | 2 | $\delta^{18}O$[‡]$, \delta D$ | Jouzel et al. (1983) |
| | | | | SP78C | 17.9** | 1.9–2.5 | $\delta D$ | Jouzel et al. (1983) |
| | | | | SP92 | 6 | 0.9–1.4 | $\delta^{18}O$ | Whitlow et al. (1992) |

* Top 3 m removed from analysis due to bad core quality.  ** No data available for first 9.6 m.
† Incomplete. Missing values linearly interpolated.  ‡ Only 0–4.9 m depth.





A diffused white noise process has the power spectral density $P_0 \exp(-\omega^2 \sigma^2)$ (van der Wel et al., 2015a), where $P_0$ is the total power of the undiffused white noise, $\sigma$ the diffusion length and $\omega$ angular frequency. The second and fourth moments are $\Omega_2 = \frac{\sqrt{\pi}}{4}\sigma^{-3}$ and $\Omega_4 = \frac{3\sqrt{\pi}}{8}\sigma^{-5}$. Thus, from Eq. (1) the average difference between two maxima is

$$\Delta z_{\text{max}} = 1/\mu_{\text{max}} = 2\pi\sqrt{\frac{2}{3}}\,\sigma\,, \qquad (2)$$

hence a linear function of the diffusion length $\sigma$ – a remarkably simple relationship.

## 2.4 Automatic estimation of the isotopic cycle length and amplitude

To investigate the isotopic variations in a similar way as visual cycle counting (Casado et al., 2016), we use an automatic procedure to identify the minima and maxima in isotope profiles. For the sake of simplicity we call the typical distance between subsequent maxima (or minima) cycle length, noting that this does not imply a periodicity that would appear as a peak in the power spectrum.

To improve robustness against measurement noise, we define a local maximum (or minimum) as that value of a profile which is above (or below) all other values within a window of $\pm 6\,\text{cm}$ centered at the given point. This naturally limits the minimum possible cycle length to $6\,\text{cm}$; however, we expect that most cycles are longer than this and the results are thus insensitive to this choice. Since we apply the same method to the observations as well as to the simulations, their intercomparison is unbiased.

We determine all local extremes for each observed or simulated profile in the described manner and record the distances between subsequent extremes (i.e., the distances between two neighboring maxima as well as between two neighboring minima) as a function of depth (midpoint of depth between the two extremes, Fig. 1). We sort the recorded distances (cycle lengths) into depth range bins, e.g. $1.5$–$2\,\text{m}$ depth, forming a distribution of distances for each specific bin. For the simulations, and also for several of our study sites, multiple profiles are available, allowing a better estimate of the cycle-length as a function of depth to be made by binning the distances from multiple profiles together. The bin width is chosen as a trade-off between maximising the resolution and minimizing the variance of the estimation. Specifically, we choose $1\,\text{m}$ for sites with one or two profiles (S2, DF, SK, MP, SP, EDML below $3.4\,\text{m}$ snow depth) and $0.5\,\text{m}$ for sites with more than two profiles (EDC, Vostok and EDML above $3.4\,\text{m}$ snow depth). For the resulting distributions, we report the mean and two standard errors ($2 \times \text{se}$). To estimate the standard error, we assume independence of the profiles except for EDML, where we assume five effective degrees of freedom as in the spectral analysis.

## 2.5 Minimal forward model for isotopic time series

As a tool to understand the observations, we construct the following minimalistic model to simulate artificial isotope profiles in the upper meters of Antarctic firn. We approximate the local climate conditions by the local near-surface air temperatures, $T_{\text{air}}(t)$, and assume that these shape the isotope signal of freshly formed snow, $\delta^{18}\text{O}_{\text{snow}}(t)$. Subsequently, the snow is transported to the surface by precipitation where it is redistributed and mixed by wind, giving rise to the surface isotope signal $\delta^{18}\text{O}_{\text{surface}}(t)$. Finally, the surface signal is buried in the firn column which is accompanied by diffusional smoothing of the signal and densification of the layers (Münch et al., 2017). Analysing a snow pit or firn core during this process then represents



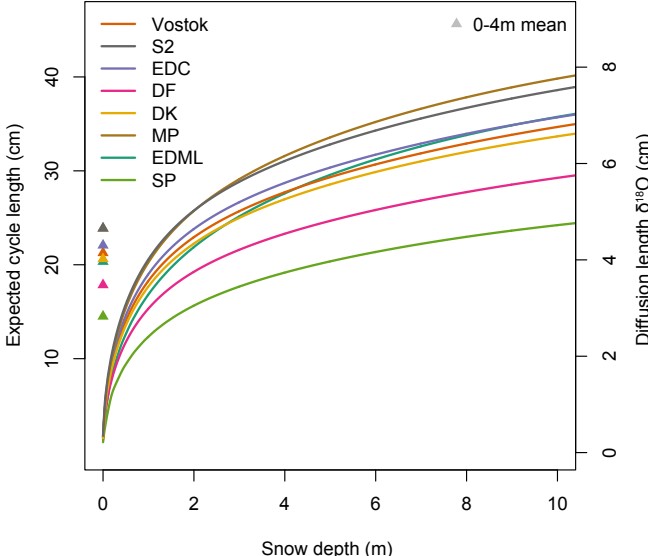

**Figure 3.** Estimated diffusion lengths for the different sites. The diffusion lengths for $\delta^{18}$O (right scale) are shown against depth. Also shown are the implied cycle lengths (left scale) assuming diffused white noise. For most sites, the mean cycle lengths in the top $4\,\mathrm{m}$ (triangles) are around $20\,\mathrm{cm}$.

a snapshot of the firn isotope signal $\delta^{18}\mathrm{O}_{\mathrm{firn}}(z)$. We describe the process here for $\delta^{18}$O but the analogous approach also applies to $\delta$D. We will now discuss these steps in detail.

On monthly to multi-decadal timescales, the local temperatures in Antarctica are dominated by the seasonal cycle. At our studied sites, the seasonal cycle explains more than $90\,\%$ of the variance in the temperature evolution of the last decades (ERA-interim reanalysis, Dee et al. (2011)) when evaluated on a monthly resolution and considering the first two harmonics of the seasonal cycle (Table 2). We therefore approximate the local climate conditions by parameterizing the seasonal cycle in temperature as

$$T_{\mathrm{air}}(t) = T_0 + A_1 \cos(\omega t + \phi_1) + A_2 \cos(2\omega t + \phi_2) + \epsilon_T. \tag{3}$$

Here, $\omega$ is angular frequency, $t$ time, $T_0$ annual mean temperature, and $A_1$, $A_2$ and $\phi_1$, $\phi_2$ denote amplitude and phase of the first two harmonics of the seasonal cycle and $\epsilon_T$ the remaining temperature variability. We estimate the parameters from temperature observations of nearby automatic weather stations (Table 2). For our study sites, $\phi_1$ and $\phi_2$ are small ($< 5°$) and are for simplicity set to zero. For converting the near-surface air temperatures into oxygen isotope ratios, we use the mean Antarctic spatial slope of $\beta = 0.8\,‰\,°\mathrm{C}^{-1}$ (Masson-Delmotte et al., 2008),

$$\delta^{18}\mathrm{O}_{\mathrm{snow}}(t) = \beta T_{\mathrm{air}}(t) + \epsilon_\delta, \tag{4}$$



where $\epsilon_\delta$ reflects the isotopic variability not captured by the linear relationship. We neglect the intercept of the calibration since the absolute isotope values have no influence on the results of our analyses and note that our results, except for the spectral comparison (Figure 8), are independent of the amplitudes and thus the choice of the calibration. A strong temporal relationship has been confirmed between the seasonal cycle of local temperature and the water isotopes measured in precipitation samples

(Eq. 4) at several sites in East Antactica (Fujita and Abe, 2006; Touzeau et al., 2016) although the estimated slopes vary from $0.3–1\%o\,°C^{-1}$ between sites.

Going from the isotope signal in the snow to the surface signal, the variability of the seasonal cycle is affected by aliasing due to precipitation intermittency (Helsen et al., 2005; Sime et al., 2009; Laepple et al., 2011; Persson et al., 2011), by redistribution of snow (Fisher et al., 1985; Münch et al., 2016; Laepple et al., 2016), and by interannual variation in the accumulation rate

and the accumulation seasonality (Cuffey and Steig, 1998). We note that exchange between atmospheric water vapor and the snow might further influence $\delta^{18}O_{surface}$ (Steen-Larsen et al., 2014; Touzeau et al., 2016; Casado et al., 2016). However, even in this case, the variations in $\delta^{18}O_{surface}$ might still follow the temperature variations. To a first approximation, precipitation intermittency, snow redistribution and interannual accumulation variability do not affect the total variance of the input signal but rather mainly redistribute its energy across frequencies, similar to the effect of aliasing (Kirchner, 2005). Thus, to simplify

matters, we describe the combination of these processes together with $\epsilon_T$ and $\epsilon_\delta$ by temporally independent ( = white) noise, and set the variance of the total surface signal (seasonal cycle + noise) to the variance of the original seasonal cycle in the snow (Eq. 4). The choice of white noise is the simplest option here and the results are actually not sensitive to this assumption (Appendix A).

Our model for the isotopic surface signal then is

$$\delta^{18}O_{surface}(t) = \beta \left( (1-\xi)^{1/2} T_{air}(t) + \xi^{1/2} \sigma_{T_{air}} \epsilon(t) \right),$$  (5)

where $\sigma_{T_{air}}$ is the standard deviation of the seasonal cycle in temperature (Eq. 3), and $\epsilon(t)$ are independent normally distributed random variables (white noise) with zero mean and standard deviation = 1. The parameter $0 \leq \xi \leq 1$ determines the fraction of noise in the surface signal: $\xi = 1$ representing the case of pure noise (completely reshuffled seasonal cycle) and $\xi = 0$ the case of a fully preserved seasonal cycle. The model also includes one implicit parameter, the resolution at which we evaluate

the variance of the noise $\epsilon(t)$. This is required as, in contrast to the seasonal cycle, white noise is not a band-limited signal. Descriptively, the parameter $\epsilon(t)$ represents the smallest spatial scale on which isotopic variations are possible. We assume 1 cm here, thus implying the complete mixing of any variations occurring on smaller spatial scales.

Finally, the burial of the surface snow transfers the surface signal time series into the depth profile $\delta^{18}O_{firn}(z)$. We approximate this process assuming a constant accumulation rate given by the present-day observations (Table 2) as the intraseasonal

and interannual variations in accumulation are already included in $\epsilon(t)$. During burial, the isotope signal is influenced by densification, layer thinning by ice flow and isotopic diffusion. Thinning of the layers is negligible in the top meters analysed here and therefore neglected in our analysis. Densification is modeled using the Herron-Langway model (Herron and Langway, 1980) assuming constant surface density and temperature for each site which are set to the modern observations (Table 2). The results are not sensitive to these simplifications since the overall effect of densification is small in the top meters of firn.



**Table 2.** Meteorological conditions and model parameters at the study sites. Listed are the annual mean temperature ($T_0$), amplitude of the first two harmonics of the seasonal cycle ($A_1$, $A_2$), annual mass accumulation rate ($\dot{b}$), firn surface density ($\rho_0$), average atmospheric pressure ($P_0$) and fraction of variance in ERA-interim monthly surface temperature explained by the seasonal cycle alone ($F_{\text{seas}}$). If borehole temperature measurements exist, we use the $10\,\text{m}$ firn temperature for $T_0$ instead of air temperature observations, as they provide a more accurate estimate of the relevant temperature for the diffusion. If no temperature observation exists, the temperature from the nearest site was used by adding the temperature anomaly between the sites from reanalysis data (Dee et al., 2011)

| Site | $T_0$ | $A_1$ | $A_2$ | $\dot{b}$ | $\rho_0$ | $P_0$ | $F_{\text{seas}}$ |
|---|---|---|---|---|---|---|---|
| | °C | °C | °C | $\text{kg m}^{-2}\,\text{yr}^{-1}$ | $\text{kg m}^{-3}$ | mbar | % |
| Vostok[a] | −55.3 | 16.7 | 6.6 | 21 | 350 | 624 | 97 |
| S2[b] | −55.1 | 17.4 | 6.3 | 21 | 350 | 642 | 97 |
| EDC[c] | −51.4 | 17.4 | 6.3 | 27 | 350 | 642 | 95 |
| DF[d] | −54.4 | 16.4 | 6.6 | 27.3 | 368 | 592 | 95 |
| DK[e] | −53.1 | 16.7 | 6.6 | 35.5 | 368 | 592 | 94 |
| MP[f] | −48.6 | 15.5 | 6.6 | 40.9 | 372 | 592 | 93 |
| EDML[g] | −44.5 | 13.2 | 4.9 | 72 | 345 | 677 | 93 |
| SP[h] | −49.9 | 15.7 | 6.4 | 83.5 | 350 | 682 | 95 |

[a] Ekaykin et al. (2002). [b] Touzeau et al. (2016), $A_1$, $A_2$ and $P_0$ from EDC.
[c] Touzeau et al. (2016). [d] Kameda et al. (2008). [e] Hoshina et al. (2014), $A_1$, $A_2$ and $P_0$
adapted from DF. [f] Hoshina et al. (2014), $A_1$, $A_2$ and $P_0$ adapted from DF.
[g] EPICA community members (2006). [h] Casey et al. (2014).

The effect of firn diffusion on the original isotope signal $\delta^{18}\text{O}_{\text{surface}}$ is modelled as the convolution with a Gaussian kernel (Johnsen et al., 2000) which leads to an overall smoothing of the input signal. The amount of smoothing is governed by the width of the convolution kernel given by the diffusion length $\sigma$, which is sensitive to ambient temperature, pressure and the density of the firn (Whillans and Grootes, 1985). We treat the dependency on density according to Gkinis et al. (2014), with

5    diffusivity after Johnsen et al. (2000). The temperature dependency of the diffusion length is highly nonlinear, with warmer temperatures leading to a stronger change. Thus, the seasonal cycle in firn temperature in the top meters increases the effective diffusion length. To approximate this effect, we follow the approach of Simonsen et al. (2011). We model the seasonal cycle in firn temperature according to the general heat transfer equation, driven by surface temperatures for constant thermal diffusivity and negligible heat advection (e.g. Paterson, 1994). We then calculate the diffusion length for parcels starting in every month of

10    the year and compute the average diffusion length over all parcel trajectories. For the sake of simplicity, we assume a constant density for the firn temperature modeling, which is set to the observed surface densities (Table 2).

The resulting effective diffusion lengths for our study sites are shown in Fig. (3). Interestingly, the combined effect of lower accumulation rates and colder temperatures largely compensate each other, leading to a rather constant diffusion length across the East Antarctic Plateau.



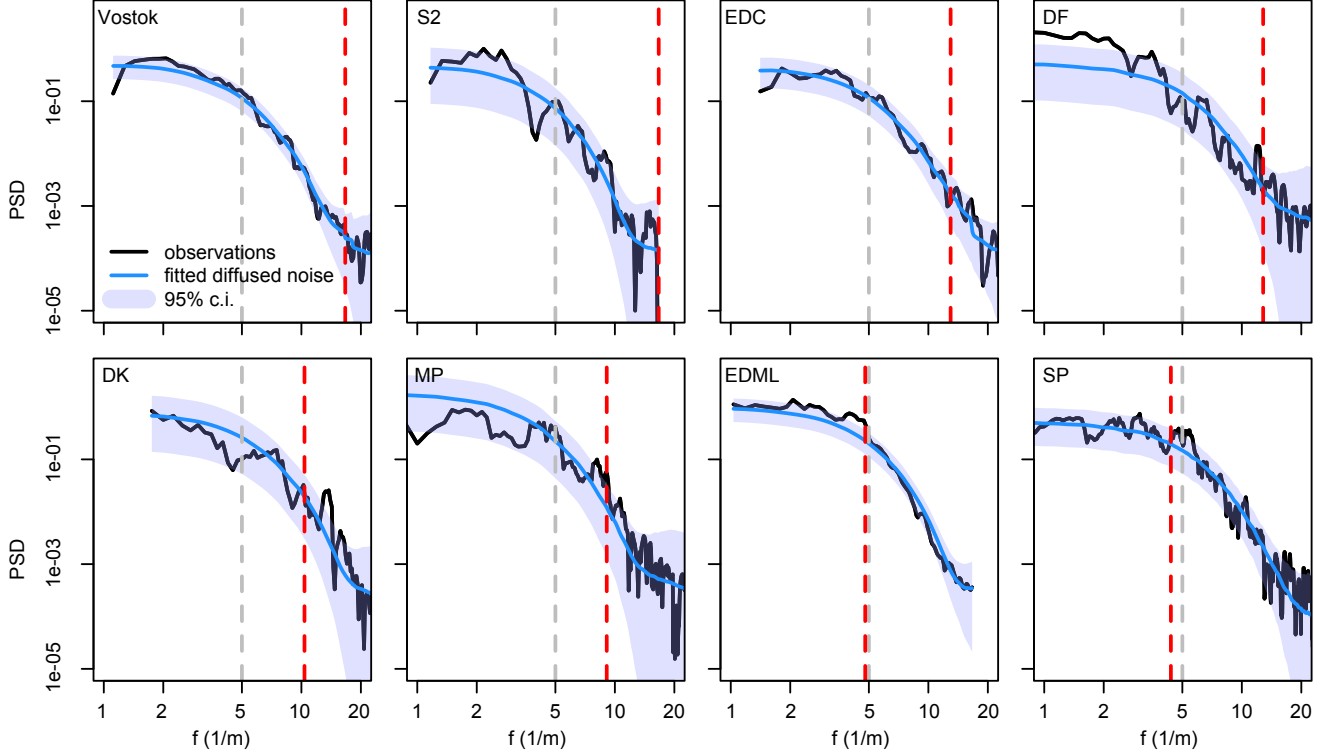

**Figure 4.** Power spectra of the firn $\delta^{18}$O variations in the firn profiles. The null hypothesis of diffused white noise (blue) and the corresponding critical significance threshold values (shaded blue) are shown. For each site, the frequency corresponding to the annual snow layer thickness is shown as a vertical red dashed line, the frequency corresponding to 20 cm is shown as a vertical grey dashed line.

## 3 Results

### 3.1 Spectral analysis of the isotope profiles

Despite originating from very different accumulation conditions, the power spectra of $\delta^{18}$O are remarkably similar across all analysed profiles (Fig. 4). The similarity of the diffused white noise null plus measurement noise spectra (blue shading) and the actual power spectra (black) suggests that the shape of the spectra are dominated by diffusion. Most sites show spectra fully consistent with diffused white noise and do not show a significant periodicity at any frequency, including frequencies corresponding to ∼20 cm (vertical grey dashed line). DF shows some significant deviation at the meter scale, corresponding to multidecadal variations. DF, MP, EDML and South Pole (SP) show locally significant peaks at frequencies close to the annual snow layer thickness (vertical red dashed line). However, even for these sites, the energy potentially related to the seasonal cycle is small, especially considering that the presumably driving temperature signal is dominated by the seasonal cycle.





## 3.2 Theoretical and observed cycle length in isotope profiles

The similarity of the power spectra between different sites, their similarity to the spectrum of diffused noise and the lack of evidence for periodic oscillations, suggests that the apparent cycles might be independent of periodic variations in the climate signal and instead represent a property of diffused noise. We note that 'noise' here just describes isotopic variations that are

largely independent in time and thus exhibit a largely time-scale invariant ('white') power-spectrum. This does neither imply a non-climatic nor negate a climatic origin of these variations. The expected cycle length for a diffused white noise signal is given by Rice's formula (Eq. 2), and equals $\sim 5$ times the diffusion length. This represents the limiting case where all initially climate-related isotopic variations would be reshuffled by precipitation intermittency and redistribution, leading to completely uncorrelated isotopic variations at the snow surface. Using the calculated diffusion lengths for the upper $4\,\mathrm{m}$ of firn gives, for

$\delta^{18}\mathrm{O}$, expected mean cycle lengths from 15 (SP) to $22\,\mathrm{cm}$ (Vostok and EDC), which are similar to those observed in isotope profiles by manual counting (Casado et al., 2016).

For a more quantitative comparison, we analyse the cycle lengths from measured $\delta^{18}\mathrm{O}$ and $\delta\mathrm{D}$ profiles using automated counting and compare them with the theoretical predictions from Rice's formula (Fig. 5). We compare here the depth range $1$–$4\,\mathrm{m}$ (or the maximum profile depth) as counting cycles in the topmost meter is more uncertain since the cycle length is a

strong function of depth. The comparison confirms the qualitative results and even shows a similarity between the variations in the observed and predicted cycle lengths (R $= 0.63$, p $= 0.06$). For most sites, $\delta^{18}\mathrm{O}$ profiles show a larger cycle length than $\delta\mathrm{D}$, which is expected since the diffusion length for $\delta^{18}\mathrm{O}$ is slightly ($\sim 10\,\%$) larger, although this is largely within the uncertainties of the estimates.

This similarity between observed cycle lengths and those predicted from diffused white noise is surprising, as we have not

yet included any climate signal, such as the seasonal cycle, in our analysis. To better understand the combined influence on the firn signal of noise, the seasonal cycle and the diffusion process, we now analyse the extent to which simulated firn profiles depend on the input signal.

## 3.3 Illustrative examples of the cycle length - depth dependency

In contrast to the diffusion length which is a function of depth, the climate signal should be largely invariant over time and thus,

in first order, independent of the depth. Therefore, investigating the depth dependency of the isotope profiles should provide us with additional insights about the origin of the variations.

To understand the depth dependency of the cycle length, as well as of the signal amplitudes, we provide three examples of simulated depth profiles (A–C) illustrating the effect of firn diffusion and noise (Fig. 6) using the environmental and depositional conditions of the EDML site (Table 2).

(A) We assume a purely periodic surface isotope signal, such as the seasonal cycle ($\xi = 0$; Fig. 6A). The cycle length, measured in snow depth units, is determined by the input signal and decreases slowly with depth due to densification. In the specific case of EDML, the cycle length is approximately $21\,\mathrm{cm}$ which is determined by the local accumulation rate of



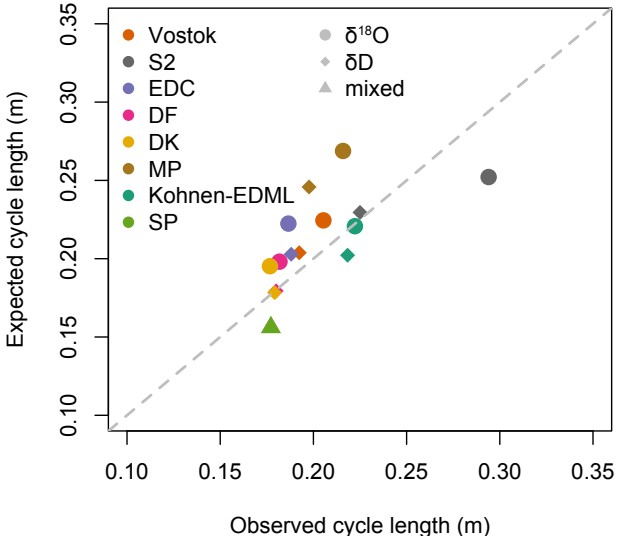

**Figure 5.** Comparison of expected and observed cycle length for all sites. The cycle length is evaluated between 1 and 4 m snow depth (or the maximum depth for shallower snow pits). Results are shown separately for $\delta D$ and $\delta^{18}O$ except for SP where a combined value is used since here $\delta D$ and $\delta^{18}O$ are not available from the same profiles. The expected cycle length is directly derived from the diffusion length assuming a pure white noise surface signal. Note that cycle lengths are larger than the one provided in Fig. 3 as the first meter is excluded from this analysis.

$\sim 72\,\mathrm{mm}$ w.eq. yr$^{-1}$ and a firn density of $345\,\mathrm{kg\,m^{-3}}$. Diffusion attenuates the initial amplitude $A_0$ of the signal with depth (Fig. 6A, middle) according to $A = A_0 \exp\left(-\frac{1}{2}\omega^2\sigma^2\right)$ (Johnsen et al., 2000).

(B) We assume that the input signal is white noise ($\xi = 1$; Fig. 6B). Possible mechanisms for such a signal would be precipitation intermittency and snow redistribution having completely reshuffled the initial seasonal signal. In this case, the expected cycle length is proportional to the diffusion length as predicted by Rice's formula and thus monotonically increases with depth. For the simulated conditions, this increase is larger than the decrease from layer thinning due to densification. The observed cycle length for a given simulation (grey dots) follows this expectation but its variability is much larger, compared to case (A), given that we now observe a stochastic instead of a deterministic periodic signal. The amplitude of the signal, as measured for example by the peak-to-peak distances, decreases near the surface and then remains largely constant. This can be heuristically understood by considering the two compensating effects: For a given cycle length or frequency $\omega$, diffusion reduces the amplitude as $\exp\left(-\frac{1}{2}\omega^2\sigma^2\right)$. For a constant cycle length, the increasing $\sigma$ with depth thus leads to an amplitude reduction. However, since the effective cycle length increases proportional to $\sigma$, both contributions cancel and lead to a constant



amplitude. For the investigated case, the cycle length again is around $20\,\mathrm{cm}$ in the top $4\,\mathrm{m}$ but here it is determined, in contrast to (A), only by the diffusion length and is not set by the input surface signal.

(C) Finally we consider a mixture of cases (A) and (B), assuming that the input signal is equally partitioned (in variance) between white noise and periodic signal ($\xi = 0.5$; Fig. 6C). In the top $\sim 10\,\mathrm{cm}$, the noise dominates and results in several

isotopic extrema. Between $\sim 10\,\mathrm{cm}$ and $\sim 6\,\mathrm{m}$, the seasonal cycle dominates the diffused signal. Then, as the amplitude of the periodic cycle is reduced, while the diffusion length increases, the diffused noise comes into play and starts to dominate the signal. The observed cycle length of isotopic maxima in this specific simulation (grey dots) thus is a mixture of cases (A) and (B), first following the annual layer thickness before transitioning to the random distances set by the properties of the diffused noise. The expected cycle length (blue line), which corresponds to the mean cycle length obtained from averaging

across multiple sites or across some meters of the profile, decreases in the top $5\,\mathrm{m}$ but then increases smoothly further down in the profile. If one counted the maxima of the isotope profile (red dots), one would reliably count the periodic signal in the upper part of the core before one would start to occasionally miss some maxima (black dots) that would otherwise be there without noise, resulting in an under-counting of the seasonal cycle in the lower part of the core.

### 3.4 Predicted and observed depth dependency of the cycle length

These examples (Fig. 6) demonstrate that very different input signals (pure noise or pure seasonal cycle) can create similar mean cycle lengths in the top meters of the firn, but that they show a distinct depth dependency. We therefore estimate cycle lengths as a function of snow depth for all our East Antarctic study sites (Fig. 7), in an attempt to distinguish between these two cases. We focus on sites for which multiple isotope profiles are available (Vostok, EDC, EDML and SP) as they allow better estimates of the variability, however, qualitatively similar results are obtained for all sites (Appendix B: Fig. 10). These results

are compared to those estimated from artificial profiles simulated for different noise fractions of the input signal, ranging from the pure seasonal cycle case ($\xi = 0$) to pure white noise ($\xi = 1$). As a reference, we additionally show the analytical result (black dashed line) for the cycle length of diffused white noise according to Rice's formula (Eq. 2).

At all sites except SP, an increase in the estimated cycle length (grey bars) is observed with depth (Fig. 7, left column). This behaviour is well reproduced by simulations that assume a high noise fraction ($\geq 50\,\%$). For all but the SP site, the observed

cycle length also follows the theoretical prediction for a pure white noise signal (black dashed line). This behaviour is in strong contrast to the cycle length in the $0\,\%$-noise case (yellow) which decreases with depth due to the thinning of the annual layer thickness by densification.

For the very low-accumulation sites, Vostok and EDC, a small noise fraction already ($10\,\%$) leads to the "diffused noise" behaviour below a depth of $0.5\,\mathrm{m}$ (Vostok), or $1\,\mathrm{m}$ (EDC), since the seasonal cycle is already strongly damped by diffusion in

the first meter. Thus, in these cases, just analysing the behaviour of the cycle length does not strongly constrain the fraction of noise vs. seasonal cycle of the surface isotope signal. In contrast, for EDML, the larger annual layer thickness and the stronger diffusion caused by the warmer temperatures lead to strongly diverging behaviour of the expected cycle lengths dependent on the ratio of noise to seasonal signal in the input. Interestingly, even at this relatively high-accumulation site, the observed cycle length increases and follows that expected and shown for simulations that assume a high noise fraction (90 or $100\,\%$).







**Figure 6.** Illustrative examples of the effect of noise and firn diffusion on the cycle length and amplitude for three input time series ($\xi = 0, 1, 0.5$). Top row: Input signal after densification. Middle row: Signal after densifiation and diffusion. The raw (2 mm resolution) data (grey) and 2 cm averages (black) mimicking a typical sampling resolution are shown. In addition, the mean expected amplitude from the Monte Carlo simulations (blue), the positions of the expected maxima of a purely periodic signal (black dots) and the positions of the actual maxima (red dots) are provided. Bottom row: Mean expected cycle length (blue) and actual distance (grey) between pairs of maxima (or minima).



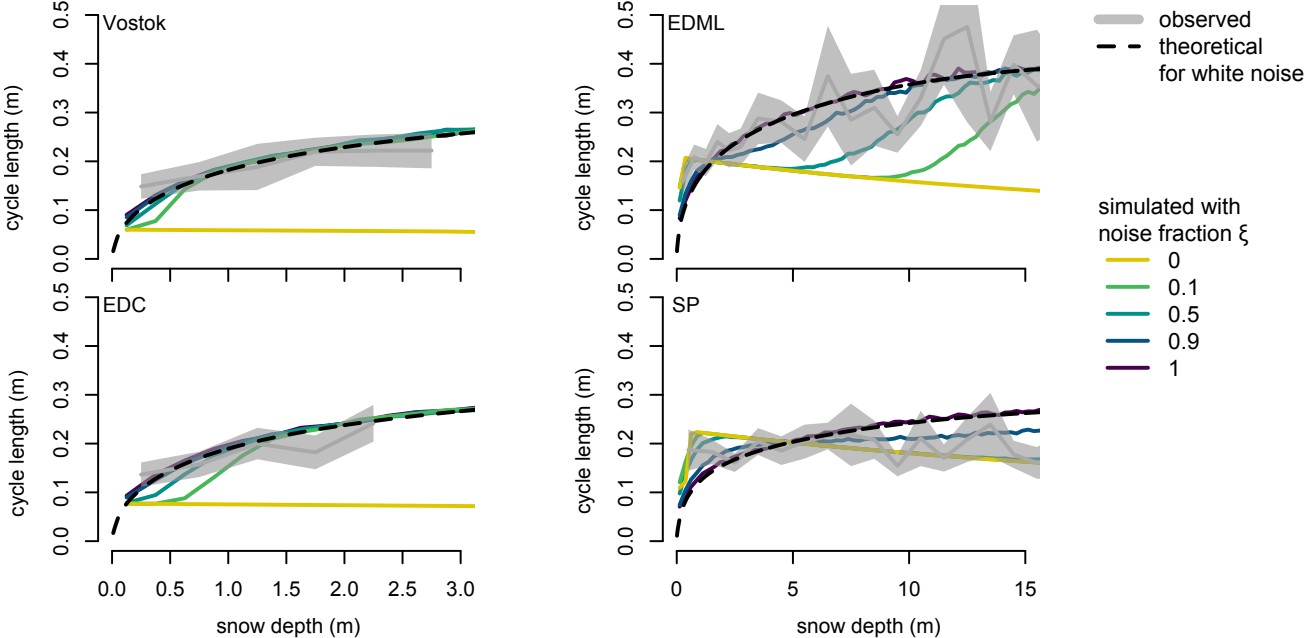

**Figure 7.** Comparison of numerically predicted and observed $\delta^{18}$O cycle statistics as a function of depth for all sites with multiple profiles. Observed cycle lengths (grey line) and a two standard error estimation uncertainty (grey band) are shown. Simulated cycle lengths are shown for various noise fractions in the input signal (coloured lines). In addition, the theoretical cycle length for pure white noise is shown (dashed black).

In contrast, a decreasing cycle length would be expected for densification of a pure, noise-free, seasonal signal. For SP, the smaller diffusion length relative to EDML results in a weaker dependency of the expected cycle lengths on the input signal. The observed cycle length is rather constant and thus lies in between the cases of 0 and 90 % noise.

5    In summary, the presented evidence suggests that, with the exception of SP, diffused noise is the dominant source of the apparent cycles at the studied sites.

### 3.5 Simulated and observed profiles and power spectra for EDML and Vostok

As a visual test of our finding, we compare the depth profiles and power spectra of simulated and observed example profiles for the two representative sites EDML (72 mm w.e. yr$^{-1}$ accumulation) and Vostok (21 mm w.e. yr$^{-1}$ accumulation) (Fig. 8). We analyse noise fractions of 5 % (seasonally dominated) and 90 % (noise-dominated). The first represents the expected case for a

10   perfect temperature proxy (perfect linear isotope-to-temperature relationship, no precipitation intermittency or redistribution, constant accumulation) as the seasonal cycle explains roughly 95 % of the total temperature variance over the last decades (Table 2). The second case represents the noise fraction that creates realistic cycle lengths and amplitudes compared to the observations (Fig. 7). For each site, a single isotope profile is shown but similar results are obtained for any of the profiles.





We first analyse the low (5 %) noise case (Fig. 8 top row). For EDML, the effect of diffusion on the amplitude of the seasonally dominated signal is moderate and only leads to a reduction in amplitude of about 50 % in the top 3.5 m. For Vostok, the annual layer thickness is much smaller and the diffusion therefore already destroys most of the seasonal signal in the top half meter, leaving only a very small diffused signal. For both sites, the diffused signal looks very much unlike the observed isotope

profiles (third row). For EDML, the diffused signal (top row) is very regular, whereas the observed profile (3rd row) shows strong interannual variations. For Vostok, the amplitude of the simulated diffused signal is much smaller than the observed amplitude of the profile.

In contrast, the high (95 %) noise cases (second row) share, for both sites, many properties with the observations. While we do not expect any correlation between the simulated and observed profiles, since the simulated profiles are by construction

largely random, the amplitude, interannual variations and cycle lengths are all similar.

These findings are confirmed by comparing the power spectra of the simulations and observations. As shown earlier (Fig. 4) the observations show a broadband spectrum with no clear periodicity. For EDML, the power spectrum of the low (5 %) noise simulation (Fig. 8 bottom row) shows clear peaks at the periods corresponding to the annual and biannual layer thickness. The broadening of the peaks arises from the varying layer thickness due to densification. In the frequency range outside the

peaks, the power spectral density, which is a measure of the time-scale dependent variance of the signal, is about one order of magnitude lower than observed. In contrast, the simulations of the high-noise case result in a power spectrum that is nearly indistinguishable from the observations. For Vostok, a similar behaviour is observed. Here, the annual peak corresponding to the layer thickness of the seasonally dominated signal is smaller, since it is strongly damped by diffusion. Again, the variance outside the peak is much too small compared to the observations, whereas the noise-dominated signal has a power spectrum

nearly indistinguishable from the observations.

## 4   Discussion and Summary

Water isotopes in firn are usually interpreted as temperature proxy. Therefore, to a first approximation, vertical isotopic variations in a snow-pit should reflect the temperature variations. The naive expectation is, therefore, that a 3 m deep profile containing 25 years and 25 seasonal cycles of climate should look very different from a 3 m profile at a higher accumulation

site that only contains 10 years of temperature variations. However, our results show that this is not the case for many sites on the East Antarctic Plateau, whose isotope profiles appear remarkably similar (Fig. 1), and this similarity is not limited to the time-series but also applies to the power spectra, which are largely indistinguishable between the sites (Fig. 4). The visual similarity of the isotopic profiles is also confirmed by systematically analysing the 'cycle length' between isotopic maxima or minima (Fig. 5).

To explain these findings, we constructed a simple forward model for isotopic signals in firn cores, similar to the ice-core proxy system model of Dee et al. (2015) or the 'virtual-ice-core model' of van der Wel et al. (2011). Our model, driven by a mixture of the seasonal cycle and white noise as input, allows the simulation of realistic isotope profiles in terms of power spectral density, amplitude and cycle length (Fig. 8). Importantly, to obtain realistic simulations of the observed firn profiles, we




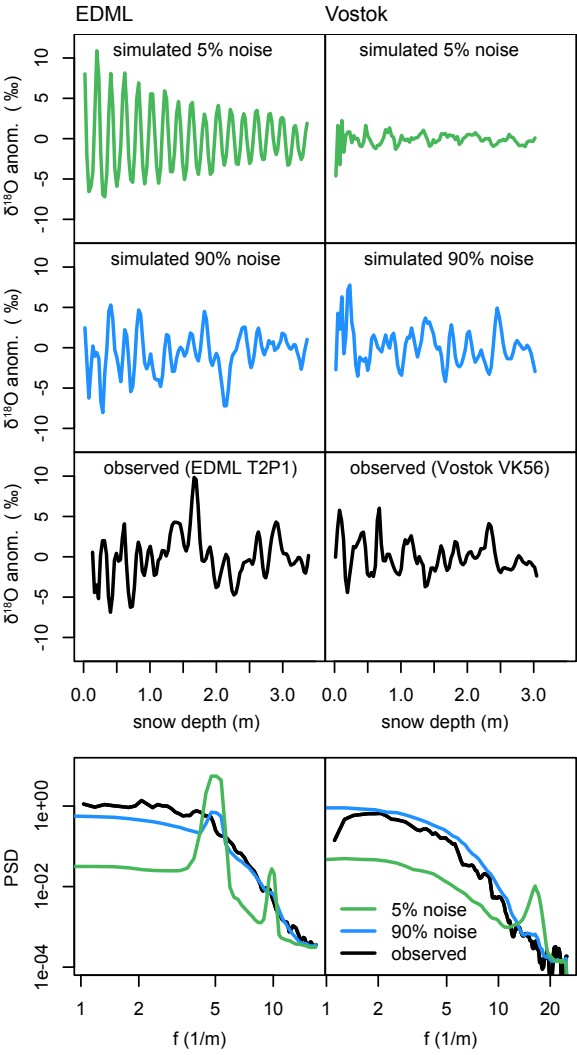

**Figure 8.** Visual comparison of measured and simulated profiles for EDML (left) and Vostok (right panel). Upper row: Simulations for 5 % noise, representing a pure climate signal. Middle row: Simulations for 95 % noise. Bottom row: Power spectra of the simulated and observed isotope profiles. The observed spectra are the mean of all spectra from the profiles available for the site. The simulated spectra are the the mean spectra over 100 realizations of the isotope profiles.





had to assume a high noise level in the input signal that represents the temporal variations of water isotopes at the surface (Fig. 7 and Fig. 8). Such a high noise level is consistent with the effect of precipitation intermittency (Helsen et al., 2005; Sime et al., 2009; Persson et al., 2011) and the stratigraphic noise caused by the redistribution of snow (Fisher et al., 1985; Laepple et al., 2016; Münch et al., 2016). Both mechanisms distort the original signal and thereby redistribute the energy from the seasonal

periodic cycle to largely uncorrelated variations. In theses cases the statistical properties, especially the power spectra of the isotope profiles, are mainly determined by isotopic diffusion, which is rather constant on the East Antarctic Plateau (Fig. 3). In turn, similar power spectra imply a similar characteristic spacing between maxima or minima, a fundamental property of stochastic processes known as Rice formula (Rice, 1944, 1945).

We applied Rice's formula to the problem of isotopic variations and showed that, assuming a white noise signal before

diffusion, the expected spacing between isotopic maxima or minima ('cycle length') is 5 times the diffusion length (Eq. 2) and thus in the order of 15-25cm in the top meters of Antarctic firn. It is important to emphasize that such a characteristic spacing of minima or maxima does not imply a periodic deterministic signal that would appear as peak in the power spectrum, as it is also the property of purely stochastic variations.

While in instrumental climate observations, a deterministic cycle (e.g. variations driven by the seasonal cycle) would be

clearly distinguishable from the realization of a purely stochastic process, this is less clear for snow-pits or firn-cores. Here, intraseasonal and interannual changes in accumulation distort the seasonal cycle (Cuffey and Steig, 1998), and might therefore smooth out potential periodic peaks in the power spectrum. However, we further showed that the depth dependency of cycle length allows discrimination between a deterministic signal (e.g. the seasonal cycle) and a stochastic signal affected by isotopic diffusion. The first case leads to a decrease in cycle length with depth, by thinning of the annual layers due to densification,

while the latter leads to an increase in cycle length with depth due to increasing diffusion length (Fig. 6, Fig. 7).

Defining noise as deviations from the surface isotopic signal that are unexplained by the local temperature time series, the depth dependency of the cycle length and amplitude suggests a significant proportion of noise in the surface isotope signal at all analysed sites (Fig. 7). While missing knowledge concerning the properties (i.e., the spectral shape) of the noise before diffusion impedes quantitative estimates, with the exception of South Pole, a noise level of 50–90% of the total variance seems

realistic for all sites analysed in this study. For South Pole, the result suggests a noise level of 10–50%.

This assertion may seem particularly troubling for the EDML site, as here the accumulation rate, as determined from snow-stakes or volcanic markers in firn cores, corresponds to an annual layer thickness of ~20cm of snow, and the cycles were usually interpreted as annual cycles (Oerter et al., 2004). In contrast, our study suggests that, at least below a depth of 3m, the isotopic 'cycles' in single profiles or cores are not dominated by the seasonal signal but rather by diffused noise (Fig. 7),

leading to the observed increase in cycle length with depth. While challenging the earlier interpretation of the variations, the new finding is consistent with the high stratigraphic noise level (50% variance) independently estimated by comparing horizontal and vertical variability in snow trenches (Münch et al., 2016), and the low reproducability between nearby firn-cores in this region (Karlöf et al., 2006).

For the three sites along the East Antarctic Divide analysed here (DF, MP, DK) and later an extended set of sites, Hoshina

et al. (2014, 2016) interpreted the multiyear cycles as the result of variable accumulation rates in combination with post-





depositional changes at the surface, such as ventilation or condensation-sublimation effects. Both studies further argued for several significant periodic cycles in partly the same firn profiles. This interpretation differs from our finding of no significant periodicities in the power-spectra of these sites (Fig. 4). In contrast to our study Hoshina et al. (2014, 2016) tested the spectral peaks against undiffused white noise. We argue that this null hypothesis will always lead to spurious significant peaks in firn

profiles (Appendix C) as the true background spectrum is modified by diffusion and is therefore not appropriate. For Vostok, Ekaykin et al. (2002) argued that the spatial dune structure results in temporal isotope variations after burial leading to the cycles. For EDC, Petit et al. (1982) discussed the potential of missing months on the isotope record, due to precipitation intermittency and erosion as well as firn diffusion, to create the structure of the isotopic variations. All four studies propose mechansims that distort or destroy the regular seasonal cycle of water isotopes and thus create noise in the isotopic record.

However, we argue that not these processes but rather the diffusion is setting the spectral structure of the signal, including the observed 'cycle length', and that this explains the similarity across sites.

At first sight, our results seem to contradict the finding that firn-core isotope profiles are significantly correlated with impurities such as $Na^+$ (Hoshina et al., 2014, 2016), especially in very low-accumulation regions. However, such a relationship is expected if the initial surface signals of isotopes and impurities are correlated, and if this correlation is not limited to high-

frequency variations. For example, if both the isotopes in snowfall and the impurities show a seasonal cycle, and both are deposited and redistributed together (i.e., wet deposition of impurities), this will result in correlated surface signals. For the typical variability of observed impurity profiles, this correlation is preserved even after diffusion (Appendix A: Fig. 9).

Our result also has implications for estimating isotopic diffusion and for the usage of layer counting. Assuming a white noise input signal, the observed cycle length is proportional to the diffusion length (Eq. 2). The agreement between the observed and

simulated cycle length (Fig. 7), and the observed spectra and diffused noise spectra (Fig. 8), thus provides some confirmation of the classical diffusion model (Johnsen et al., 2000). Recently is has been argued that the diffusion model of Johnsen et al. (2000) overestimates diffusion length in the top meters of the firn, at least in Greenland (van der Wel et al., 2015b). Focusing on the same dataset might potentially allow this to be formally tested but this is beyond the scope of this study.

We showed that the level of noise in the input signal also determines the depth dependency of the amplitude of the variations.

The boundary case of a diffused pure seasonal cycle leads to an exponential decrease of amplitude with depth (Johnsen et al., 2000), whereas a diffused white noise signal results in a slower decrease of the the peak to peak amplitude (Fig. 6, Fig. 7). All isotope signals in snow or firn will be noise affected, due to stratigraphic noise (Fisher et al., 1985; Laepple et al., 2016; Münch et al., 2016) and precipitation intermittency (Helsen et al., 2005; Sime et al., 2009; Persson et al., 2011). Therefore, estimates of diffusion strength based on analysing the decay of the seasonal cycle amplitude by measuring the peak to peak

amplitude in the time domain (Cuffey and Steig, 1998) might be biased low if they do not account for the noise. Related to this issue, our result also underlines the fact that layer counting in isotope profiles should only be performed after undiffusing the isotope signal (Cuffey and Steig, 1998), or using non-diffused parameters such as impurities. Our results suggest that layers in the deeper parts of the firn could be systematically missed by simply counting the local extremes in isotope profiles (Fig. 6), leading to age models that are biased towards 'younger' ages.



The combination of isotopic diffusion with strong variability at the surface that is not directly related to temperature, also limits the effective resolution of climate signals that can be obtained by analysing water isotopes. While the problem of diffusion could be overcome by undiffusing the signal (Cuffey and Steig, 1998), this procedure also inflates the noise. Therefore, methods to reduce the noise by averaging across cores (Münch et al., 2016), or the use of other parameters, have to be employed

when aiming for high-resolution climate reconstructions at low accumulation sites.

## 5    Conclusions

We provide an explanation of why snow pits across different sites in East Antarctica show visually similar variations in water isotopes. We argue that the similar power spectra and apparent cycles of around 20 cm in near surface isotope profiles are the result of a seasonal cycle in isotopes, noise, for example from precipitation intermittency, and diffusion. The near constancy of

the diffusion length across many ice-coring sites (Fig. 3) explains why the structure and cycle length is largely independent of the accumulation conditions. At some sites, such as EDML, the cycle length implied by the isotopic diffusion coincides with the annual snow layer thickness in the upper meters of the firn. This calls for a careful consideration of the effects of noise and diffusion when interpreting isotopic variations.

Our hypothesis does not exclude the existence of a climatic signal in the isotope timeseries, as any low-frequency surface

signal would still be preserved in the diffusion process, and thus does not question the relevance of water isotopes as a paleo-temperature proxy. However, for low accumulation areas in particular, we show that the typical spacing of extrema in isotopic profiles can be explained without invoking multi-decadal climate changes or other climate related hypotheses.

Our results underline previous findings that water isotope signals in low-accumulation regions have a small signal-to-noise ratio. Therefore, methods to reduce the noise such as averaging across cores have to be employed when aiming for high-

resolution climate reconstructions. Finally, systematically analysing the spectral shape of isotopic variability and and not just the potential periodicities and cycles might be a promising way forward to quantitivly understand the isotopic variability in polar firn cores (Fig. 8).

## 6    Data availability

The water isotope data will be made available in PANGAEA after acceptance.

## 7    Appendix A: Sensitivity to the input signal

Our previous calculations assumed an isotopic surface signal that is a mixture of a seasonal cycle and uncorrelated (white) noise. While uncorrelated noise is the simplest hypothesis, it is likely that the surface signal exhibits more structure. Potential processes that lead to autocorrelation include precipitation events that deposit several centimeters of snow with similar isotopic composition, as well as mixing and redistribution by wind drift that might vertically homogenize the snow surface.





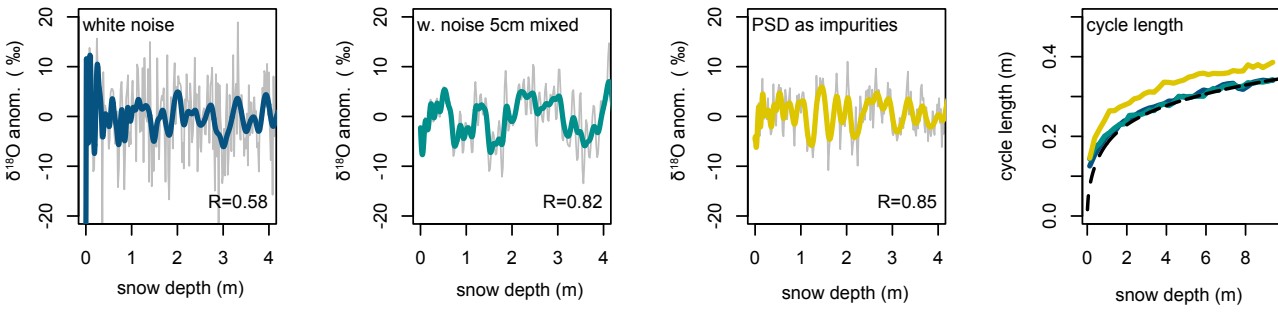

**Figure 9.** Sensitivity of cycle length to the temporal correlation structure of the assumed input signal. Thin lines show the undiffused signal, thick coloured lines the signal after diffusion. For the more structured signals (mixed white noise and variability mimicking impurities), a larger fraction of the signal is preserved, leading to higher correlations. The resulting cycle length (right panel, coloured lines) is only weakly dependent on the input signal and is close to the theoretical result for white noise (Rice's formula, dashed line).

In contrast to the isotopic composition, major ions preserved in snow and firn are not affected by diffusion. Assuming an atmospheric source, wet deposited impurities are also influenced by precipitation intermittency and snow redistribution and might therefore provide information about the correlation structure of the signal in the surface snow. Interestingly, major ion profiles in snow-pits (e.g. Hoshina et al., 2014, 2016) are clearly distinguishable from white noise. This suggests that at least
5   some correlated structure is preserved or created in the depositional process.

To test the effect of autocorrelated noise in the input signal on the resulting cycle length, we simulate profiles assuming three different input signals: white noise, white noise subject to a $5\,\mathrm{cm}$ mixing (low-pass filtered with a finite response filter with cutoff frequency $1/10\,\mathrm{m}^{-1}$), and noise constructed with a similar temporal structure as observed impurity profiles. For the latter, we estimate the mean power spectrum of the $4\,\mathrm{m}$ long $Na^+$ impurity profiles of DF and MP (Hoshina et al., 2014),
10   and generate new random time series from this spectrum.

The results (Fig. 9) show that although the input signal strongly differs, the diffused signal is very similar. The resulting cycle lengths for the white noise and the mixed white noise inputs are identical, and both close to the theoretical expectation (dashed line). The cycle length of the impurity-based simulation is slightly higher ($\sim 3\,\mathrm{cm}$ offset).

We note that while the cycle length is similar, the correlation between the input and the diffused signal is larger for the more
15   structured input signals as a larger fraction of low-frequency variability is preserved after diffusion.



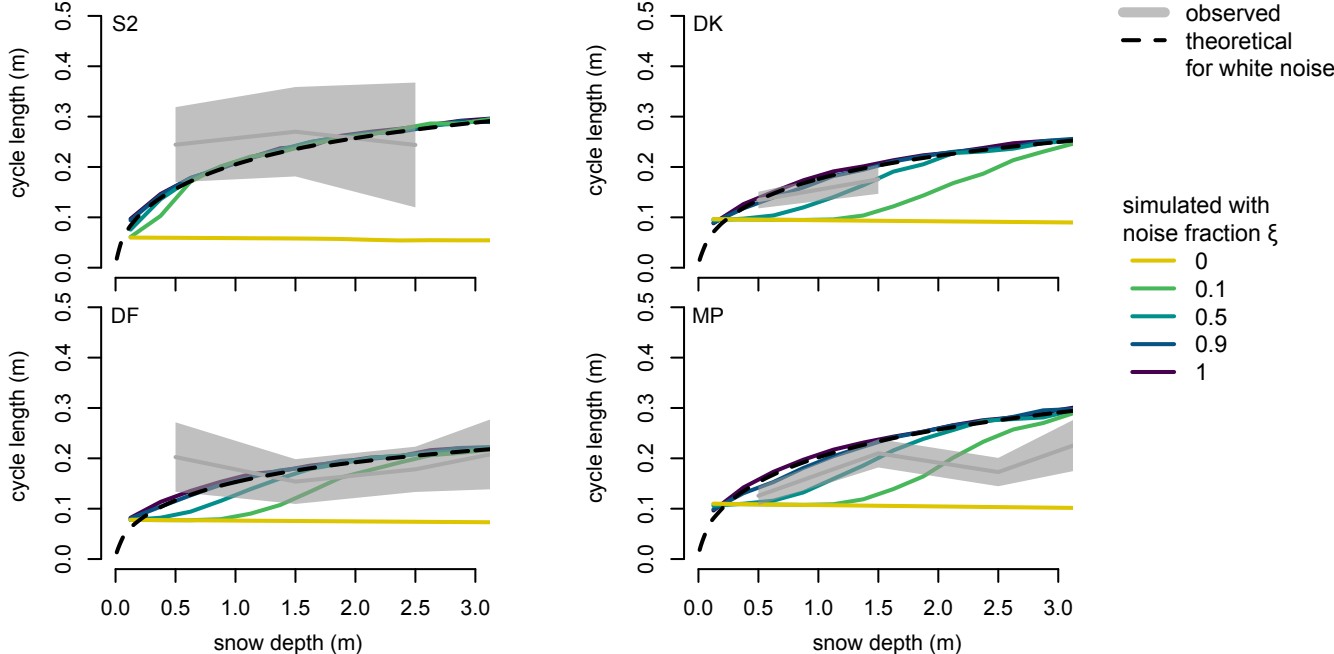

**Figure 10.** As (Fig. 7) but for the remaining sites which only have a single isotopic firn profile. Observed cycle lengths (grey line) and a two standard error estimation uncertainty (grey band are shown). Simulated cycle lengths are shown for various noise fractions in the input signal (coloured lines). In addition the theoretical cycle length for pure white noise is shown (dashed black).

## 8 Appendix B: Observed and simulated cycle length for the remaining profiles

In the main text, we showed the observed and simulated cycle length statistics for the sites with multiple profiles (Fig. 7), as they allow a better estimation of the cycle length. The four remaining sites with single profiles (Fig. 10) also show cycle lengths consistent with the high noise level simulations.

5  The depth dependency of the cycle length is less clear which is likely caused by the large estimation uncertainty. In addition, MP shows a systematically smaller observed cycle length than the simulations. Potential reasons could be either uncertainties in the isotopic dataset (independent noise leads to more minima and maxima and thus a smaller cycle length) or our choice of climatic parameters (accumulation rate, firn temperature).

## 9 Appendix C: Spurious significance when using a white noise null hypothesis

10  To demonstrate the effect of a white noise null hypothesis on the spectral analysis of water isotopes, we simulate random $\delta^{18}O$ profiles using our minimal forward model. To mimic Hoshina et al. (2014), we use the site parameters for DF and a pure white noise ($\xi = 1$) input signal that is subject to the site specific densification and diffusion. The final data are averaged to 3 cm to



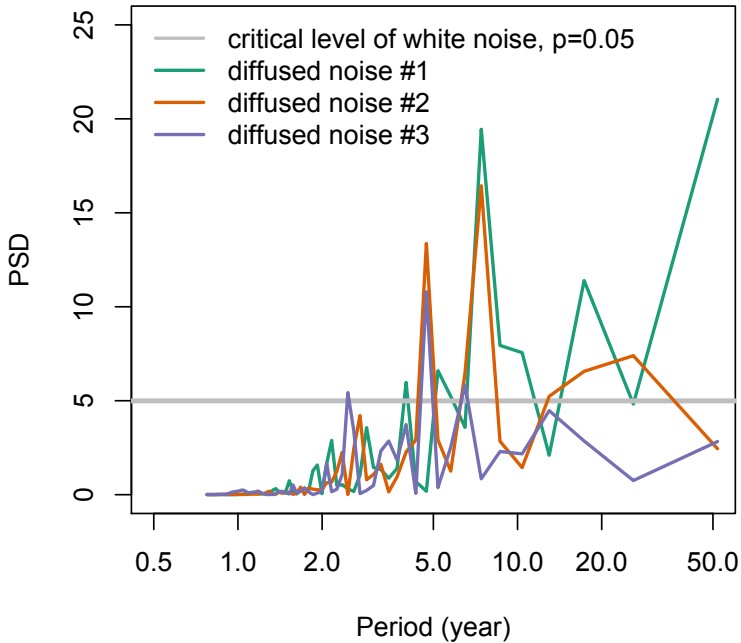

**Figure 11.** Demonstration of spurious peaks when testing against a white noise null hypothesis. Coloured lines show power spectra for three realizations of purely random firn profiles. The grey horizontal line shows the critical level (p=0.05) that has to be overcome to be locally significant against white noise. As the simulated firn profiles are subject to firn diffusion, and thus have a red power spectra, they show random peaks higher than the critical level.

mimic a typical sampling interval We estimate the power spectrum using a raw periodogram and show the p=0.05 significance level of a white noise null hypothesis. For all three realizations of purely random firn profiles, the spectra show energy well above the white noise significance level (Fig. 11). This demonstrates the need for using a null hypothesis that accounts for the isotopic diffusion.

5 **10 Competing interests**

The authors declare that they have no conflict of interest





## 11 Acknowledgments

We thank Y.Hoshina and A. Ekyakin for sharing their data and J. Freitag and A M. Dolman for discussions and detailed editing. T.L. and T.M. were supported by the Initiative and Networking Fund of the Helmholtz Association Grant VG-NH900. M.C and A.L. have received funding from the European Research Council under the European Union's Seventh Framework Programme (FP7/2007-2013) / RC grant agreement number 306045.



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
