# Peer review of "On the similarity and apparent cycles of isotopic variations in East Antarctic snow-pits"

_The Cryosphere, 2017_

## Referee Comment (RC1) · B. Markle (Referee) · 17 Oct 2017

General Comments

In this paper the authors investigate an interesting observation that water isotope profiles in shallow firn cores and snow pits show rather constant peak-to-peak wavelengths across East Antarctic sites despite very different accumulation rates. The authors suggest that the source of largely uniform water-isotope cycle length is water isotope diffusion acting on a mostly uncorrelated white noise background, which leads to a characteristic wavelength of water isotope excursions that is linear with the diffusion length. Similar diffusion lengths across East Antarctica thus leads to the similar cycle lengths observed.

[Figure]

This is an excellent paper. The problem is well laid out in generally clear language. The study is very well designed. In particular I found the build up of the minimal forward model and the comparison between observations and synthetic model based expectations to be excellent and informative. The figures are clear and attractive. I found the arguments in this study to be compelling.

There is an increasing body of work on the diffusion of water isotopes in firn and the interpretation of such records, largely in the spectral domain. This study takes insights from that body of work to inform traditional time-domain interpretations of water-isotope records in snow and firn. This study provides a much-need caution to the community in interpreting "annual", interannual, and perhaps decadal variability from these records - in particular single records from the high plateau. It will make a useful and novel contribution to the field.

Generally the paper is well written though both the abstract and introduction have some confusing phrasing or language. There were also some minor errors throughout.

Specific Comments and questions: I have a couple larger questions and a few minor ones.

1) From Equation 2 and the diffusion lengths estimated in Figure 3, it appears that the expected peak to peak distances ought to increase quite substantially in even shallow firn cores (10m depth or so), perhaps up to double the 20cm discussed throughout the text. Figure 7 shows this quite nicely for a couple of cores up to about 15 m. But certainly by the bottom of the firn the wavelength of peaks ought to be substantially longer, be well developed, and presumably locked in. Is such an increase in peak-to-peak distance observed in deeper firn cores or indeed the main deep ice cores from these sites? Such a prediction should be easily verifiable. Is there any other processes not discussed that would alter such characteristic spacing of isotope peaks at greater depths? Perhaps firn compaction and eventually thinning will begin to take over? But it seems that an increased in the peak-to-peak distance in the 20 to 50 m depth range is

a prediction of this work? Can this be verified?

2) On page 20 (line 17) you sate that the correlation of impurities and water isotopes should survive the diffusion process. I completely agree with the arguments about the redistribution of both isotopes and impurities. But the impurities are not subject to the same diffusion process (as you quite nicely discuss in the appendix). Are you arguing that the correlation survives the diffusion of just the water isotope record? If so, how much (if any) is that initial correlation degraded with increasing diffusion? If initially correlated water isotope and impurity records are both similarly whitened by redistribution (or whatever process) and then the water isotopes are smoothed by diffusion, the impurities ought to have higher power high frequency content than the water isotopes. Is this observed? Further the impurities ought to have very little seasonal cycle surviving (at least for some sites since you find the water isotopes to be best represented by a high initial noise fraction per Figure 7). Is this observed?

I also don't really understand why in Appendix A and Figure 9 you apparently diffused the impurity-like record. As you state, the impurities are not subject to diffusion. Perhaps I have misunderstood what was done for that particular profile, if so the description on page 22 could be a little more clear.

The following is an observation rather than a question. An underlying assumption in water isotope diffusion based studies is that the initial background spectrum of the variability is white noise (or close to it and with the possible notable exception of the annual cycle). Your results, in particular those in Figure 7, seem to be a nice confirmation of this. Your Appendix A seems to suggest that (at least some) autocorrelation in the initial signal is possible though doesn't change the resultant signal much.

Minor comments, questions:

Figure 1 is nice. But it would be much more convincing to see several of these example time series (even just in a supplementary figure) than just the two shown here.

Section 2.2: Line 10: "temporally independent" – so no annual cycle? (This seems like a good null hypothesis, if so.) Line 16: Is the local diffusion length not also a free parameter? Or is that independantly estimated? Section 2.3: This was fascinating, thanks. Section 2.4: Line 13: How does 6cm compare to the minimum accumulation rate at a site you investigated? This is worth mentioning.

Sec 2.5 (page 4), Line 13: This is a fine first assumption or starting place. However was this slope not estimated from data that may contain seasonal or multi-seasonal averages as well as data from depth, and thus including the post depositional processes of interest here? I'd be curious if the final slope of T_air d18O_snow, after your forward model, is the same as this initial value. This is mostly just curiosity; this does not appear to influence your results in any meaningful way.

Figure 6 is excellent and informative.

Technical Corrections:

Abstract:

Line 4: "depict" is likely the wrong word. Perhaps "record".

Line 5: Reorder sentence for clarity. ". . . variation in accumulation rate between sites". Also the meaning of the ending of the sentence is not very clear: "matched by variation in the number of seasonal cycles". Presumably you mean the number of seasonal cycles for a given depth.

Line 6: "Accumulation conditions" is vague. Do you mean "accumulation rates"?

Line 15: It's confusing to put "Finally" in the middle of the paragraph. (Though the line makes an excellent point!)

Line 17: "similar power" is two words.

Section 1: (Page 2) Line 10: "which extent" should be "what extent". Line 17: "the the"

Line 22: Is "annual snow layer thickness" the same as annual accumulation rate? If so this might be useful to state.

Section 2.4 (page 7) Line 6: Reword for clarity: "...in a way similar to..."

Pg 12, Line 25: The depth dependency of what about the isotope profiles?

Pg 20, line 21: Missing "it". Should be "...it has been argued..."

---

## Referee Comment (RC2) · Anonymous Referee #2 · 17 Oct 2017

"On the similarity and apparent cycles of isotopic variations in East Antarctic snow-pits"

This manuscript explores cycles in stable water isotopes in the upper layers of snow across East Antarctica, and their relationship to temperature changes: examining the non-temperature 'noise' signal, and impact of diffusion, on cycle length. I like this paper: the comparison between predicted and observed cycle lengths; the attempt to assess how much of the signal is seasonal/periodic versus white noise; the use of both frequency analysis and layer counting type techniques. The manuscript is thoughtfully put together and well written.

General points:

It is not always clear whether the authors are talking about sub-annual signals when

considering signal-to-noise ratios, or something else – such as annual mean signals.

The sites considered here are 'low-accumulation' sites: low accumulation should be clarified. Further, it would very helpful if the analysis could be extended beyond low-accumulation sites to mid-accumulation (and perhaps high accumulation) Antarctic sites: sub-annual climate/temperature signals have never been retrieved from low accumulation sites. Thus the value of this work might actually be in looking at where and when sub-annual temperature information CAN be gathered from stable water isotope measurements from snow pits. This would imply that the authors should look at the question of what is NOT a low-accumulation (too noisy) site.

It is interesting that the signal to noise ratio is perhaps in the region of 90-10% for low accumulation sites but again, what is it for other higher accumulation sites?

It is less clear what is the impact of this work beyond understanding snow pits / firn cores. Are these findings relevant to the deeper East Antarctic ice cores?

Specific points:

P1 L4 "isotopes in surface snow are supposed. . ." rephrase to make clear that isotopes in snow are really not intrinsically supposed to do anything.

L17 similarpower

L21 clarify what low-accumulation means – e.g. less than 80 mm weq per year. It would be worth clarifying through the introduction.

P4 Figure 2. What are the colours for the stations/core sites? And would it not be more useful to plot accumulation, rather than temperature?

P7 L12, it would intuitively seem more likely that a +16 cm smoothing would lead to a minimum cycle length of 12 cm. Clarify/explain please?

L28 Could condensation temperature be used instead – more common to relate d18O to condensation, rather than surface temperature.

P9 L17 remove "actually"

---

## Referee Comment (RC3) · Anonymous Referee #3 · 25 Oct 2017

The manuscript is devoted to the study of the nature of the high-frequency (with periods < 1 m) cycles observed in the vertical profiles of stable isotopes in snow-firn thickness of central Antarctica. The authors convincingly show that these cycles with a typical wavelength of the order of 20 cm could be formed solely by the diffusion of the initially random (i.e. consisted of white noise) values of snow isotopic content, thus confirming a very low signal-to-noise ratio in the isotopic series.

To my opinion, this manuscript is an important new step towards understanding the process of formation of the vertical isotopic profiles in Antarctic snow, which, in turn, crucial for correct interpretation of the deep ice core isotopic data.

I do not have major remarks for this manuscript, only a few minor comments and corrections, as listed below.

[Figure]

A general note: in your model of isotopic vertical profiles you consider the diffusive smoothing but neglect another part of this process, namely, isotopic and mass exchange with the atmosphere that may lead to the alteration of the mean isotopic composition of snow (compared to that of precipitation). I may guess that this process does not play role here, but it would be better to state it explicitly.

The title: probably it worth thinking of a title that better describes the main result of the manuscript, e.g.: "Non-climatic origin of the apparent cycles in (high-frequency?) stable water isotope variability in central Antarctic snow"... It would look stronger and more attractive.

Page 2 lines 3-5 – may be better to write "The ice thickness and accumulation rate affects the temporal scale and resolution of the climate reconstructions that can be obtained from a given ice core"

Page 2 line 16 – "including those at Vostok..."

Section 2.5 title – better write "vertical profiles" instead of "time series"

Page 9 line 6 – better cite Stenni et al (2016) where a good review of different slopes is given:

https://www.the-cryosphere.net/10/2415/2016/tc-10-2415-2016.pdf

Table 2 – if 10 m temperature is preferable, then it is better to take -57*C for Vostok instead of -55*C (Lefebvre E., L. Arnaud, A. Ekaykin, V.Y. Lipenkov, G. Picard and J.R. Petit. - Snow temperature measurements at Vostok station from an autonomous recording system (TAUTO): preliminary results from the first year operation. - Ice and snow, 2012, v. 4, p. 138-145.)

Figure 8 caption – should be "simulation for 90% noise"?

Page 20 lines 10-11 – just a comment: I see your point that the observed cycles can be solely explained by the suggested mechanism (white noise + diffusion), so you do not

need to invoke other processes. But this does not necessarily exclude the influence of dunes that may be hidden somewhere in this noise, as the large error bars on your spectra allow it. Moreover, at least on the scale of mega-dunes it is proven that the dunes do influence the spatial and temporal variability of the snow stable water isotope content (https://www.the-cryosphere.net/10/1217/2016/tc-10-1217-2016.pdf), so the dunes do matter. However, the influence of the dunes on the relevant time scales (years or decades) is still an open question.

Page 20 line 26 – an odd "the"

Page 21 line 20 – an odd "and"

Title of Appendix B – I suggest to rename to "Observed and simulated cycle length for the sites with one (single) available profile"

Page 24 line 1 – missed dot.

Page 25 line 2 – please replace "Ekyakin" by "Ekaykin"

---

## Author Comment (AC1) · 12 Nov 2017

We would like to thank all three reviewers for the constructive and detailed comments. We respond below, repeating the original question first, before each answer or comment is given in blue and the proposed changes in the revised version in red.

**Reviewer 1 (B. Markle)**

General Comments

In this paper the authors investigate an interesting observation that water isotope profiles in shallow firn cores and snow pits show rather constant peak-to-peak wavelengths across East Antarctic sites despite very different accumulation rates. The authors suggest that the source of largely uniform water-isotope cycle length is water isotope diffusion acting on a mostly uncorrelated white noise background, which leads to a characteristic wavelength of water isotope excursions that is linear with the diffusion length. Similar diffusion lengths across East Antarctica thus leads to the similar cycle lengths observed.

This is an excellent paper. The problem is well laid out in generally clear language. The study is very well designed. In particular I found the build up of the minimal forward model and the comparison between observations and synthetic model based expectations to be excellent and informative. The figures are clear and attractive. I found the arguments in this study to be compelling.

There is an increasing body of work on the diffusion of water isotopes in firn and the interpretation of such records, largely in the spectral domain. This study takes insights from that body of work to inform traditional time-domain interpretations of water-isotope records in snow and firn. This study provides a much-need caution to the community in interpreting "annual", interannual, and perhaps decadal variability from these records - in particular single records from the high plateau. It will make a useful and novel contribution to the field.

Generally the paper is well written though both the abstract and introduction have some confusing phrasing or language. There were also some minor errors throughout.

Specific Comments and questions: I have a couple larger questions and a few minor ones.

1) From Equation 2 and the diffusion lengths estimated in Figure 3, it appears that the expected peak to peak distances ought to increase quite substantially in even shallow firn cores (10m depth or so), perhaps up to double the 20cm discussed throughout the text. Figure 7 shows this quite nicely for a couple of cores up to about 15 m. But certainly by the bottom of the firn the wavelength of peaks ought to be substantially longer, be well developed, and presumably locked in. Is such an increase in peak-to- peak distance observed in deeper firn cores or indeed the main deep ice cores from these sites? Such a prediction should be easily verifiable. Is there any other processes not discussed that would alter such characteristic spacing of isotope peaks at greater depths? Perhaps firn compaction and eventually thinning will begin to take over? But it seems that an increased in the peak-to-peak distance in the 20 to 50 m depth range is a prediction of this work? Can this be verified?

Thanks for this suggestion. The increase in the upper firn is proportional to the time elapsed since deposition. This increase is however counteracted in the deeper firn by the decrease in diffusivity due to the higher firn densities, and by the decrease in annual layer thicknesses due to the densification and, later, the thinning by ice flow (neglecting the diffusion in solid ice). Together, this leads to the specific depth behaviour of the diffusion length (e.g. Figure 2 in (Johnsen et al., 2000)) and thus also of the cycle length.

Analysing deeper firn cores suggests that our model is also valid deeper down (Review Figure 1). This match is surprising as our simple model omits thinning, and as the quality of these firn-profiles on the high-frequency side is unclear; since e.g. the sampling resolution varies from 1 to 10cm. Further it is unclear if the white noise assumption holds also deeper in the firn, where the same depth interval corresponds to a longer time period and thus might potentially include a stronger climate signal.

[Figure]

*Review Figure 1: Comparison of the cycle lengths of d18O (larger dots) from the firn cores B34 and B37 drilled near EDML and the theoretical cycle length assuming a white noise signal. Also shown is the varying sampling resolution (small circles) of the cores. The difference in cycle lengths between B34 and B37 and the changes of the variability of the cycle length estimations over depth are likely related to the changes in sampling resolution.*

Given that a quantitative analysis (accounting for thinning, changes in the sampling resolution, detailed evaluation of the data quality of these partly unpublished cores) is beyond the scope of this paper, we would prefer to leave this analysis out of the revised manuscript but are open to include the qualitative analysis (as Review Figure 1) if suggested by the editor.

2) On page 20 (line 17) you state that the correlation of impurities and water isotopes should survive the diffusion process. I completely agree with the arguments about the redistribution of both isotopes and impurities. But the impurities are not subject to the same diffusion process (as you quite nicely discuss in the appendix). Are you arguing that the correlation survives the diffusion of just the water isotope record? If so, how much (if any) is that initial correlation degraded with increasing diffusion?

Yes, we argue that the initial correlation should partly survive just the diffusion of the isotope record. The degradation of the initial correlation will depend on the initial power spectra of the isotopes/impurities and on the dependency of the initial correlation on frequency (the cross-spectrum). In the panels of Figure 9 we provide the correlations calculated in the 4m window assuming 1.) the initial power spectrum as indicated (white, w. with 5cm mixed…) and 2.) an initial correlation of 1 with no time-scale dependency.

*In simple words; if there is a correlation of the slower variations not affected by diffusion this part will survive the diffusion process.*

We will rephrase the statement to clarify that the correlation is "partly preserved" and improve the figure caption of Fig. 9.

If initially correlated water isotope and impurity records are both similarly whitened by redistribution (or whatever process) and then the water isotopes are smoothed by diffusion, the impurities ought to have higher power high frequency content than the water isotopes. Is this observed? Further the impurities ought to have very little seasonal cycle surviving (at least for some sites since you find the water isotopes to be best represented by a high initial noise fraction per Figure 7). Is this observed?

*Indeed, for the snow-pits with impurity as well as oxygen isotope data we find more high frequency content in the impurity than in the water isotope data (Review Figure 2)*

[Figure]

*Review Figure 2: Power spectra of the firn Na+ and d18O variations; Shown are the mean spectra from DF & MP (Hoshina et al., 2014) after normalizing every time-series to unit variance. It is clearly visible that Na+ contains more high frequency variability than d18O.*

*For very low accumulation regions (<50mm w.eq./year) the impurities seem to have no or little seasonal cycle surviving (Hoshina et al., 2014, 2016). For the EDML region this is less clear. Sommer et al., (2000) found a seasonal cycle e.g. in Na+ usable for counting while our study argues for a noise level of 50- 90% variance (evaluated on the cm scale). This suggests that either we are more on the lower side of the noise (e.g. 50%), or the original signal in the EDML region might have been different for impurities and d18O with more seasonality in the impurities.*

We will add a discussion of these points in the revised manuscript

I also don't really understand why in Appendix A and Figure 9 you apparently diffused the impurity-like record. As you state, the impurities are not subject to diffusion. Per- haps I have misunderstood what was done for that particular profile, if so the description on page 22 could be a little more clear.

The aim of Appendix A is to test the sensitivity of our hypothesis to the spectral structure of the input signal. As we don't know the original / surface variability of water isotopes since what we observe has been already affected by diffusion, we use the power spectrum of the impurities as a possible guess how the power spectrum of the water isotopes might have been before diffusion.
We will improve the description and motivation of this analysis in the Appendix

The following is an observation rather than a question. An underlying assumption in water isotope diffusion based studies is that the initial background spectrum of the variability is white noise (or close to it and with the possible notable exception of the annual cycle). Your results, in particular those in Figure 7, seem to be a nice confirmation of this. Your Appendix A seems to suggest that (at least some) autocorrelation in the initial signal is possible though doesn't change the resultant signal much.

Yes, our results suggest that at least in the frequency range corresponding to snow depth ranges of 1 m to several cm, the original spectrum of isotopic variability is close to white; This is also confirmed by Figure 4 as diffused white noise (using the independently estimated diffusion length) can explain most of the observed power spectra. On the very fast side (e.g. 2-10cm) we don't have a good constraint as uncertainties in the diffusion or the measurement noise have a stronger effect, and the effect on the cycle length of changes in variations on the very fast side (Appendix A) are small.

We will add a sentence in the discussion that our results support the assumption of an initially white spectrum that is used in water isotope based diffusion studies.

**Minor comments, questions:**

Figure 1 is nice. But it would be much more convincing to see several of these example time series (even just in a supplementary figure) than just the two shown here.

We generally agree with the reviewer on this point; However, our study was motivated by a partner study that proposed the existence of the 20cm cycles and showed several example time-series from the same core-set (Figure 12 in Casado, M., A. Landais, G. Picard, T. Münch, T. Laepple, B. Stenni, G. Dreossi, et al. 2017. "Archival Processes of the Water Stable Isotope Signal in East Antarctic Ice Cores." The Cryosphere Discuss. 2017 (November):1–36. https://doi.org/10.5194/tc-2017-243. (formerly: https://www.the-cryosphere-discuss.net/tc-2016-263/). The original plan was that this partner study appears first and we follow up with this quantitative analysis. Now the order is reversed but we would still avoid to show a very similar figure.
In the revised version, we will point the reader to the figure in the partner manuscript.

Section 2.2:
Line 10: "temporally independent" – so no annual cycle? (This seems like a good null hypothesis, if so.)

Yes, we assume no annual cycle. As we also write in the previous sentence "Significance testing of the power spectral density is performed against a null hypothesis of noise affected by firn diffusion" and do not mention any annual cycle in this section, we think that this is clearly defined and didn't change the text.

Line 16: Is the local diffusion length not also a free parameter? Or is that independently estimated?

We use an independent estimate by modelling the local diffusion length based on the site-specific parameters of temperature, accumulation and surface density. The null hypothesis is therefore largely independent of the observed spectra. We clarified this by changing the sentence to "The isotopic diffusion length is calculated using the site-specific temperature, accumulation and density as described in Section 2.5.

Section 2.3: This was fascinating, thanks.

Section 2.4: Line 13: How does 6cm compare to the minimum accumulation rate at a site you investigated? This is worth mentioning.

The 6 cm equals the minimum annual layer thickness (= snow accumulation) that we observe at Vostok and S2. We changed the sentence to "*however, as the accumulation rates at our sites vary from 6-21cm snow per year, we expect that most cycles are longer than this and the results are thus insensitive to this choice.*"

Sec 2.5 (page 4), Line 13: This is a fine first assumption or starting place. However was this slope not estimated from data that may contain seasonal or multi-seasonal averages as well as data from depth, and thus including the post depositional processes of interest here? I'd be curious if the final slope of T_air d18O_snow, after your forward model, is the same as this initial value. This is mostly just curiosity; this does not appear to influence your results in any meaningful way.

Good point. Spatial slopes should not be affected as the processes considered here (diffusion & densification) do not affect the mean. Temporal slopes are of course strongly affected by the deposition processes with noise (e.g. by redistribution or intermittency) as well as diffusion tending to reduce the slope. This is one reason why we cite and use the spatial slope.

Figure 6 is excellent and informative.

**Technical Corrections:**

Abstract:
Line 4: "depict" is likely the wrong word. Perhaps "record".

Changed to "record".

Line 5: Reorder sentence for clarity. ". . . variation in accumulation rate between sites". Also the meaning of the ending of the sentence is not very clear: "matched by variation in the number of seasonal cycles". Presumably you mean the number of seasonal cycles for a given depth.

We will rephrase this sentence in the revised version.

Line 6: "Accumulation conditions" is vague. Do you mean "accumulation rates"?
We mean more than just the accumulation rates as other processes that have been used to describe the cycles (e.g. dunes, postdepositional effects…) not just depend on the accumulation rate but also on the windspeed etc.

Line 15: It's confusing to put "Finally" in the middle of the paragraph. (Though the line makes an excellent point!)

We will rephrase this sentence in the revised version.

Line 17: "similar power" is two words.
Corrected

Section 1: (Page 2) Line 10: "which extent" should be "what extent".
Corrected

Line 17: "the the"
Corrected

Line 22: Is "annual snow layer thickness" the same as annual accumulation rate? If so this might be useful to state.

Yes, it is the same; We changed it to "Interestingly, despite the very different accumulation rates at these sites, which range from 6 to 21cm snow per year"

Section 2.4 (page 7) Line 6: Reword for clarity: ". . .in a way similar to. . ."
Changed as proposed

Pg 12, Line 25: The depth dependency of what about the isotope profiles?
Changed to "Therefore, investigating the depth dependency of the cycle length in isotopic profiles"

Pg 20, line 21: Missing "it". Should be ". . .it has been argued. . ."
Corrected

**Anonymous Referee #2**

"On the similarity and apparent cycles of isotopic variations in East Antarctic snow-pits"
This manuscript explores cycles in stable water isotopes in the upper layers of snow across East Antarctica, and their relationship to temperature changes: examining the non-temperature 'noise' signal, and impact of diffusion, on cycle length. I like this paper: the comparison between predicted and observed cycle lengths; the attempt to assess how much of the signal is seasonal/periodic versus white noise; the use of both frequency analysis and layer counting type techniques. The manuscript is thoughtfully put together and well written.

**General points:**
It is not always clear whether the authors are talking about sub-annual signals when considering signal-to-noise ratios, or something else – such as annual mean signals.

We define the noise fraction $\xi$ on Page 9, Line 21-27 as the variability fraction unrelated to the seasonal cycle and evaluated on the 1cm scale. It therefore corresponds to the variance integrated from sub-annual signals until decadal or multidecadal (the length of the snow-pits). In the result section and figures, we always refer to the defined $\xi$. In the discussion, we will clarify the meaning of the noise: "*This noise fraction is estimated from snow-pits of several meters and is evaluated on the cm scale; It therefore corresponds to sub-annual to decadal variations*"

The sites considered here are 'low-accumulation' sites: low accumulation should be clarified.
We agree that these are low-accumulation sites. We tried to clearly define the accumulation range of the analysed sites. In the Abstract we clearly define the range of accumulation: 20 to 80mm w.e.,yr^-1. We now modified the discussion to emphasize again the accumulation range and defined low-accumulation: "…*However, our results show that this is not the case for many low-accumulation (<100 mm w.e.) sites on the Antarctic Plateau, whose isotope profiles appear remarkably similar*

Further, it would very helpful if the analysis could be extended beyond low- accumulation sites to mid-accumulation (and perhaps high accumulation) Antarctic sites: sub-annual climate/temperature signals have never been retrieved from low accumulation sites.

We generally agree that it would be worthwhile to extend such a study to other sites with higher accumulation or even including Greenland. Preliminary studies (including NEEM in Greenland) indeed show that our hypothesis is applicable beyond low accumulation sites. However, we feel that such an extension is beyond the scope of this study.

Thus the value of this work might actually be in looking at where and when sub-annual temperature information CAN be gathered from stable water isotope measurements from snow pits. This would imply that the authors should look at the question of what is NOT a low-accumulation (too noisy) site. It is interesting that the signal to noise ratio is perhaps in the region of 90-10% for low accumulation sites but again, what is it for other higher accumulation sites?

While we see our study as one step towards a general understanding of the isotopic variability and signal to noise ratios, we are not (yet) able to provide a comprehensive estimate of the signal to noise ratio as a function of accumulation (and time-scale). We are aiming for such an estimate but this will need further detailed studies. With that in mind, we would refrain from widening the scope of the present work at this point.

It is less clear what is the impact of this work beyond understanding snow pits / firn cores. Are these findings relevant to the deeper East Antarctic ice cores?

Indeed, the main focus of our study is on snow and firn. However, our findings form a basis for the understanding of the signal formation that is essential for the interpretation water isotopes in all firn or ice profiles. It will further also affect the variations seen in the ice below the firn-ice transition (Review Figure 1).

Specific points:
P1 L4 "isotopes in surface snow are supposed. . ." rephrase to make clear that isotopes in snow are really not intrinsically supposed to do anything.
We rephrased this line in the Abstract

L17 similarpower
Corrected

L21 clarify what low-accumulation means – e.g. less than 80 mm weq per year. It would be worth clarifying through the introduction.
We defined the accumulation rates of the analysed sites in the Abstract and also now define "low-accumulation" in the conclusions (*However, our results show that this is not the case for many low-accumulation (<100 mm w.e.) sites on the Antarctic Plateau, whose isotope profiles appear remarkably similar*).
However, in general, we do not see any evidence for a clear threshold for the mechanisms of the signal formation or the signal content depending on a specific accumulation rate and therefore try not to introduce or emphasize any 'artificial threshold'.

P4 Figure 2. What are the colours for the stations/core sites? And would it not be more useful to plot accumulation, rather than temperature?
Thanks for spotting this; We will remove the colours. We would prefer to keep the temperature as show variable as 1.) to our knowledge there are no reliable gridded accumulation estimates covering the whole of Antarctica and 2.) temperature is equally important for diffusion as accumulation.

P7 L12, it would intuitively seem more likely that a +-6 cm smoothing would lead to a minimum cycle length of 12 cm. Clarify/explain please?

We do not smooth over +- 6cm but in the search for a maximum (minimum), we do not allow a second maximum (minimum) inside a distance of smaller than 6cm. Therefore, the minimal distance between two maxima (or minima) is 6cm leading to a minimum cycle length of 6cm. We will clarify this in the method description.

L28 Could condensation temperature be used instead – more common to relate d18O to condensation, rather than surface temperature.
Yes, condensation temperature could be used instead of surface temperature without any change of the results or conclusions, as our results to not depend on the absolute scaling of the seasonal cycle

or the calibration, and condensation temperature and surface temperature are strongly correlated on the time-scales of interest. Given that both is possible, we decided to use surface temperature for simplicity as 1.) we can rely on data from the automatic weather stations instead of having to rely on climate models or reanalysis , 2.) it is often used as reconstruction target e.g. (Jouzel et al., 1997).

P9 L17 remove "actually"
Removed

**Anonymous Referee #3**

The manuscript is devoted to the study of the nature of the high-frequency (with periods < 1 m) cycles observed in the vertical profiles of stable isotopes in snow-firn thickness of central Antarctica. The authors convincingly show that these cycles with a typical wavelength of the order of 20 cm could be formed solely by the diffusion of the initially random (i.e. consisted of white noise) values of snow isotopic content, thus confirming a very low signal-to-noise ratio in the isotopic series.
To my opinion, this manuscript is an important new step towards understanding the process of formation of the vertical isotopic profiles in Antarctic snow, which, in turn, crucial for correct interpretation of the deep ice core isotopic data.
I do not have major remarks for this manuscript, only a few minor comments and corrections, as listed below.

A general note: in your model of isotopic vertical profiles you consider the diffusive smoothing but neglect another part of this process, namely, isotopic and mass exchange with the atmosphere that may lead to the alteration of the mean isotopic composition of snow (compared to that of precipitation). I may guess that this process does not play role here, but it would be better to state it explicitly.

Thanks for this comment; Yes, we limit ourselves by purpose on the classical processes that provide the "null hypothesis" for the signal. We now state this explicitly when describing our model:
We note that we limit our model to the known first order processes and do not consider isotopic and mass exchange with the atmosphere that may potentially lead to an alteration of the mean isotopic composition of snow (Casado et al., 2017; Steen-Larsen et al., 2014).

The title: probably it worth thinking of a title that better describes the main result of the manuscript, e.g.: "Non-climatic origin of the apparent cycles in (high-frequency?) stable water isotope variability in central Antarctic snow". . . It would look stronger and more attractive.
Thank you for this suggestion. While we agree that our title is very 'general' we feel that it also describes our broad approach (spectral analysis, forward modelling, theoretical model) on the isotopic variations. We also have difficulties with the proposed title as diffusion is also controlled by climate and it includes two poorly defined terms ("high frequency", "central Antarctic").

Page 2 lines 3-5 – may be better to write "The ice thickness and accumulation rate affects the temporal scale and resolution of the climate reconstructions that can be obtained from a given ice core"
We changed the sentence as proposed.

Page 2 line 16 – "including those at Vostok. . ."
Changed as proposed

Section 2.5 title – better write "vertical profiles" instead of "time series"
Changed as proposed

Page 9 line 6 – better cite Stenni et al (2016) where a good review of different slopes is given: https://www.the-cryosphere.net/10/2415/2016/tc-10-2415-2016.pdf
We added Stenni et al., (2016) but also left Fujita and Abe (2006) and Touzeau et al., (2016) as they show the strong relationship between temperature and the isotopic composition of precipitation.

Table 2 – if 10 m temperature is preferable, then it is better to take -57*C for Vostok instead of -55*C (Lefebvre E., L. Arnaud, A. Ekaykin, V.Y. Lipenkov, G. Picard and J.R. Petit. - Snow temperature measurements at Vostok station from an autonomous recording system (TAUTO): preliminary results from the first year operation. - Ice and snow, 2012, v. 4, p. 138-145.)
Thanks. In the calculations, we already used -57*C (as this is also the value in Touzeau et al.,). We now corrected the table and added the reference to Lefebvre et al.,

Figure 8 caption – should be "simulation for 90% noise"?
Corrected.

Page 20 lines 10-11 – just a comment: I see your point that the observed cycles can be solely explained by the suggested mechanism (white noise + diffusion), so you do not need to invoke other processes. But this does not necessarily exclude the influence of dunes that may be hidden somewhere in this noise, as the large error bars on your spectra allow it. Moreover, at least on the scale of mega-dunes it is proven that the dunes do influence the spatial and temporal variability of the snow stable water isotope content (https://www.the-cryosphere.net/10/1217/2016/tc-10-1217-2016.pdf), so the dunes do matter. However, the influence of the dunes on the relevant time scales (years or decades) is still an open question.
We fully agree with the reviewer. In the previous sentence to the one in question, we also mention that for example dunes are one effect that distorts the signal = creates the noise. Further our mechanism provides a sufficient mechanism but does not rule out other influences or explanations. We changed the sentence to "*setting the first order spectral structure*"

Page 20 line 26 – an odd "the" Page 21 line 20 – an odd "and"
Corrected

Title of Appendix B – I suggest to rename to "Observed and simulated cycle length for the sites with one (single) available profile"
Changed as suggested

Page 24 line 1 – missed dot.
Corrected

Page 25 line 2 – please replace "Ekyakin" by "Ekaykin"
Corrected

**References**

Casado, M., Landais, A., Picard, G., Münch, T., Laepple, T., Stenni, B., … Jouzel, J. (2017). Archival processes of the water stable isotope signal in East Antarctic ice cores. *The Cryosphere Discuss.*, *2017*, 1–36. https://doi.org/10.5194/tc-2017-243

Hoshina, Y., Fujita, K., Nakazawa, F., Iizuka, Y., Miyake, T., Hirabayashi, M., … Motoyama, H. (2014). Effect of accumulation rate on water stable isotopes of near-surface snow in inland Antarctica. *Journal of Geophysical Research: Atmospheres*, *119*(1), 274–283. https://doi.org/10.1002/2013JD020771

Hoshina, Y., Fujita, K., Iizuka, Y., & Motoyama, H. (2016). Inconsistent relationships between major ions and water stable isotopes in Antarctic snow under different accumulation environments. *Polar Science*, *10*(1), 1–10. https://doi.org/10.1016/j.polar.2015.12.003

Johnsen, S. J., Clausen, H. B., Cuffey, K. M., Hoffmann, G., Schwander, J., & Creyts, T. (2000). Diffusion of stable isotopes in polar firn and ice: the isotope effect in firn diffusion. *Physics of Ice Core Records*, *159*, 121–140.

Jouzel, J., Alley, R. B., Cuffey, K. M., Dansgaard, W., Grootes, P., Hoffmann, G., … others. (1997). Validity of the temperature reconstruction from water isotopes in ice cores. *Journal of Geophysical Research*, *102*(C12), 26471–26487. https://doi.org/10.1029/97JC01283

Sommer, S., Appenzeller, C., Röthlisberger, R., Hutterli, M. A., Stauffer, B., Wagenbach, D., … Mulvaney, R. (2000). Glacio-chemical study spanning the past 2 kyr on three ice cores from Dronning Maud Land, Antarctica: 1. Annually resolved accumulation rates. *Journal of Geophysical Research: Atmospheres*, *105*(D24), 29411–29421. https://doi.org/10.1029/2000JD900449

Steen-Larsen, H. C., Masson-Delmotte, V., Hirabayashi, M., Winkler, R., Satow, K., Prié, F., … Sveinbjörnsdottir, A. E. (2014). What controls the isotopic composition of Greenland surface snow? *Climate of the Past*, *10*(1), 377–392. https://doi.org/10.5194/cp-10-377-2014